# Genetic epidemiology of dengue viruses in phase III trials of the CYD tetravalent dengue vaccine and implications for efficacy

Maia A Rabaa[1,2]*, Yves Girerd-Chambaz[3], Kien Duong Thi Hue[1], Trung Vu Tuan[1], Bridget Wills[1,2], Matthew Bonaparte[4], Diane van der Vliet[3], Edith Langevin[3], Margarita Cortes[5], Betzana Zambrano[6], Corinne Dunod[3], Anh Wartel-Tram[7], Nicholas Jackson[3], Cameron P Simmons[1,2,8]*

[1]Oxford University Clinical Research Unit, Hospital for Tropical Diseases, Ho Chi Minh City, Vietnam; [2]Centre for Tropical Medicine, Nuffield Department of Medicine, University of Oxford, Oxford, United Kingdom; [3]Research and Development, Sanofi Pasteur, Lyon, France; [4]Global Clinical Immunology, Sanofi Pasteur, Swiftwater, United States; [5]Research and Development, Sanofi Pasteur, Bogota, Colombia; [6]Research and Development, Sanofi Pasteur, Montevideo, Uruguay; [7]Medical Affairs and Public Policy Asia AP, Sanofi Pasteur, Singapore, Singapore; [8]Department of Microbiology and Immunology, Peter Doherty Institute, University of Melbourne, Victoria, Australia

*For correspondence: mrabaa@oucru.org (MAR); csimmons@unimelb.edu.au (CPS)

**Abstract** This study defined the genetic epidemiology of dengue viruses (DENV) in two pivotal phase III trials of the tetravalent dengue vaccine, CYD-TDV, and thereby enabled virus genotype-specific estimates of vaccine efficacy (VE). Envelope gene sequences (n = 661) from 11 DENV genotypes in 10 endemic countries provided a contemporaneous global snapshot of DENV population genetics and revealed high amino acid identity between the E genes of vaccine strains and wild-type viruses from trial participants, including at epitope sites targeted by virus neutralising human monoclonal antibodies. *Post-hoc* analysis of all CYD14/15 trial participants revealed a statistically significant genotype-level VE association within DENV-4, where efficacy was lowest against genotype I. In subgroup analysis of trial participants age 9–16 years, VE estimates appeared more balanced within each serotype, suggesting that genotype-level heterogeneity may be limited in older children. Post-licensure surveillance is needed to monitor vaccine performance against the backdrop of DENV sequence diversity and evolution.
DOI: https://doi.org/10.7554/eLife.24196.001

## Introduction

Dengue is the commonest arboviral disease of humans and has been a major public health problem in tropical Asia and Latin America for decades (*Stanaway et al., 2016*). Reducing the population of competent mosquito vectors of dengue viruses has been the central aim of disease control efforts, but these have had little success in eliminating or stopping the spread of dengue globally. Effective dengue vaccines will be essential tools to achieving dengue control. Accordingly, the licensure of the first tetravalent dengue vaccine (chimeric yellow fever–dengue virus tetravalent dengue vaccine (CYD-TDV), Sanofi Pasteur) together with recommendations from The World Health Organisation's Strategic Advisory Group of Experts (SAGE) on Immunization on its use in highly endemic countries,

**eLife digest** Each year, about 100 million people—mostly children in tropical parts of Asia and Latin America—are infected with the dengue virus. It has been difficult to produce a vaccine against the virus, because there are four different types of the virus, and people respond to infections with different types in an unusual way. Once a person is infected with one type of dengue, they are protected from future infections with that type. However, if that person later becomes infected with a different type, they are more likely to experience severe illness. As a result, a dengue vaccine must simultaneously protect against all four types of the virus to be safe and effective.

The first dengue vaccine has recently become available. Clinical studies of the vaccine show that it can protect against all four virus types, but that the protection against certain types and in some age groups varies. Complicating matters, the four types of the dengue virus have continued to evolve since scientists first began developing the vaccine. Therefore, scientists are concerned that the vaccine may not be as effective against the newly evolved subtypes.

To find out, scientists would have to carefully compare the genetics of the strains used to develop the vaccine with the strains currently circulating. They would also have to see how well the vaccine protects against current strains.

Now, Rabaa et al. show that there is a high level of genetic similarity between the viruses used to create the vaccine, and dengue viruses that caused infections in people participating in clinical studies of the vaccines. The analyses also showed that in children between the ages of 2 and 16, the vaccine is more effective against one subtype of the dengue type-4, compared to the other circulating subtype. In children between the ages of 9 and 16, who are eligible to receive the vaccine in some countries, the vaccine was largely equally effective across the various subtypes.

In addition to providing reassurance that the vaccine is working against currently circulating types, Rabaa et al. provide a valuable snapshot of the genetic diversity of dengue viruses. This snapshot will help scientists develop more effective dengue vaccines and treatments. More studies following vaccinated people are needed to ensure that the current vaccine remains effective as circulating strains of the virus evolve.

DOI: https://doi.org/10.7554/eLife.24196.002

has provided the first prospects of an integrated public health approach to disease control (*WHO, 2016*).

Dengue vaccine development has been challenging, in part because dengue viruses (DENV) exist as four phylogenetically and antigenically distinct serotypes (DENV-1 to −4). Within each virus serotype exists considerable genetic diversity at local, national and continental scales (*Holmes, 2008*). Subtle antigenic differences can also be measured amongst members of the same virus serotype and are speculated to be of epidemiological and clinical importance (*Katzelnick et al., 2015*). The virus population dynamics of DENV in hyperendemic areas are complex, often involving the emergence and extinction of viral lineages against a backdrop of multiple virus types co-circulating and oscillating in their relative prevalence. Human population immunity and intrinsic virus fitness in mosquitoes and humans are potential drivers of DENV evolution in these settings (*Holmes and Burch, 2000*). Acting to balance high mutation rates of DENV within individual hosts, the vector-human transmission cycle subjects viral populations to strong purifying selection, whereby emergent virus variants that are less fit for disseminated infection of both humans and mosquitoes are lost from the viral population (*Holmes, 2003*).

CYD-TDV was found to be safe and efficacious for use in children 9 years of age and older, with efficacy varying according to age, baseline serostatus and virus serotype (*Capeding et al., 2014*; *Villar et al., 2015*). Furthermore, a trend toward reduced efficacy against DENV-2 was observed in the Asian phase III trial compared to the Latin American trial (*Hadinegoro et al., 2015*). This finding suggested that the efficacy of CYD-TDV might be affected by sub-serotype (i.e. genotype) level diversity in DENV populations, often associated with geographical boundaries. Beyond the epidemiological factors identified in previous studies of CYD-TDV efficacy, the performance of dengue vaccines could also be influenced by the evolving nature of DENV populations in endemic settings. For example, the possibility that circulating DENV populations could 'escape' vaccine-elicited immune

responses was nominated as one of several possible explanations for the relatively low efficacy of CYD-TDV against DENV-2 in a phase IIb trial in Thailand (*Sabchareon et al., 2012*). Two phase III efficacy trials of CYD-TDV, involving more than 31,000 children between the ages of 2–14 years in the Asia–Pacific region (CYD14 trial) and between the ages of 9–16 years in Latin America (CYD15 trial) (*Hadinegoro et al., 2015*) enable, for the first time, a *post hoc* investigation of vaccine efficacy versus DENV population diversity. Thus, the aims of the present study were threefold. First, to document the genetic distance between the components of the CYD-TDV formulation and the DENV strains detected amongst cases in the CYD14 and CYD15 trials. Second, to perform focused analysis of the level of sequence conservation between CYD-TDV vaccine strains and wild-type DENV at epitope locations targeted by potent virus neutralising human monoclonal antibodies (mAbs). Lastly, we aimed to explore if a more complex genotype-specific efficacy pattern existed in the CYD14 and CYD15 trials, notwithstanding the limitations inherent to *post hoc* analysis. Collectively, these data provide insights into the characteristics of the CYD-TDV product relative to contemporary DENV populations and provide preliminary insight into genotype-level vaccine efficacy that can serve as a baseline for future research.

## Results

### Acquisition of DENV envelope gene sequences

433 acute serum samples from 595 virologically-confirmed dengue (VCD) cases in CYD14 and 512 samples from 662 VCD cases in CYD15 were selected for investigation on the basis of subject consent, viremia level and sample volume considerations (*Figure 1A and B*, respectively). From CYD14, 314 complete DENV envelope (E) gene nucleotide sequences (1485 nt for DENV-1,–2, −4; 1479 nt for DENV-3) were acquired directly from 433 serum samples (72.5%, including three mixed infections), with a subset of 299/433 (69.1%) samples also having a complete premembrane (prM) nucleotide sequence. From CYD15, 333 complete DENV E gene nucleotide sequences were acquired directly from 512 serum samples (65.0%, including eight mixed infections), with a subset of 313/512 (61.1%) samples also having a complete prM nucleotide sequence. The proportion of serum samples that yielded an E gene sequence was similar between control and dengue vaccine recipients within each study (*Supplementary file 1a*). The probability of acquiring an E gene sequence from serum samples was positively associated with the DENV viremia level (*Figure 1—figure supplement 1*).

### Phylogenetic profile of CYD-TDV vaccine strains and DENV detected in CYD14 and CYD15 trials

Full and partial E gene sequences determined directly from serum samples collected in CYD14 and CYD15 trials (253 DENV-1, 191 DENV-2, 107 DENV-3 and 110 DENV-4) were aligned with E gene sequences corresponding to the CYD-TDV vaccine strains and sequences from GenBank for which the year and country of sampling were known. Maximum likelihood trees representing subsampled E gene sequence datasets allowed the classification of CYD14/15 viruses to the major intra-serotype lineages (genotypes) previously described for DENV (*Figure 2—figure supplements 1–4*). At the country level, CYD14/15 viruses were closely related to publicly available DENV sequences acquired from the same country, an indicator of ongoing local evolution. *Figure 2* shows the genotypes detected in the CYD14/15 virus populations according to their country of sampling. Collectively, these data define the population genetics of viruses responsible for dengue cases in the CYD14/15 trials and provide a unique contemporaneous snapshot of DENV diversity in ten endemic countries.

### Sequence differences between CYD-TDV vaccine strains and circulating wild-type viruses

We quantified the differences between the E gene amino acid sequences in the components of the tetravalent CYD-TDV formulation and viruses from VCD cases in the CYD14 and CYD15 trials. The mean level of E gene amino acid sequence difference between vaccine strains and viruses from VCD cases in CYD14 and CYD15 was <3% for all serotypes (*Figure 3* and *Supplementary file 1b*). To define the nature of these sequence differences, the amino acid positions that varied between CYD-TDV vaccine strains and the E gene sequences sampled in CYD14/15 trials and in the subsampled GenBank sequences were annotated adjacent to the subsampled maximum likelihood phylogenetic

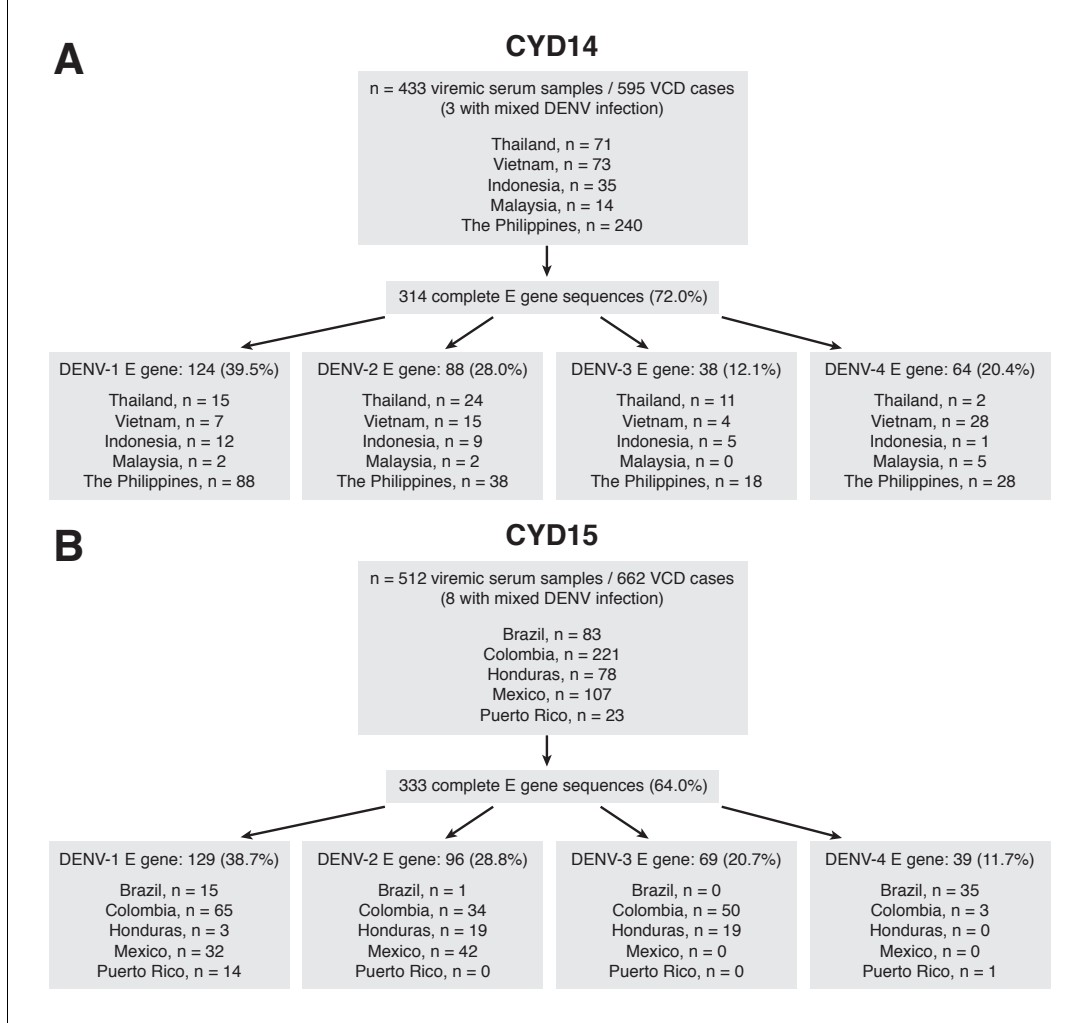

**Figure 1.** Sequencing flow chart for samples obtained in CYD-TDV trials. (**A**) CYD14, (**B**) CYD15.

DOI: https://doi.org/10.7554/eLife.24196.003

The following source data and figure supplement are available for figure 1:

**Source data 1.** Sequence alignment of DENV-1 prM and E genes from CYD-TDV trials.

DOI: https://doi.org/10.7554/eLife.24196.005

**Source data 2.** Sequence alignment of DENV-2 prM and E genes from CYD-TDV trials.

DOI: https://doi.org/10.7554/eLife.24196.006

**Source data 3.** Sequence alignment of DENV-3 prM and E genes from CYD-TDV trials.

DOI: https://doi.org/10.7554/eLife.24196.007

**Source data 4.** Sequence alignment of DENV-4 prM and E genes from CYD-TDV trials.

DOI: https://doi.org/10.7554/eLife.24196.008

**Figure supplement 1.** Probability of sequencing success versus viremia.

DOI: https://doi.org/10.7554/eLife.24196.004

trees for each serotype. The DENV-2 E gene phylogeny (incorporating the vaccine strain) of relevance to the CYD14 trial is shown in *Figure 4A* and for CYD15 in *Figure 4B*. The equivalent annotated phylogenies for DENV-1,–3 and −4 are shown in *Figure 4—figure supplements 1–6*. These data reveal that positions of amino acid non-identity between CYD-TDV vaccine strains and wild-type viruses were dispersed across the E protein and do not cluster to any particular structural domain.

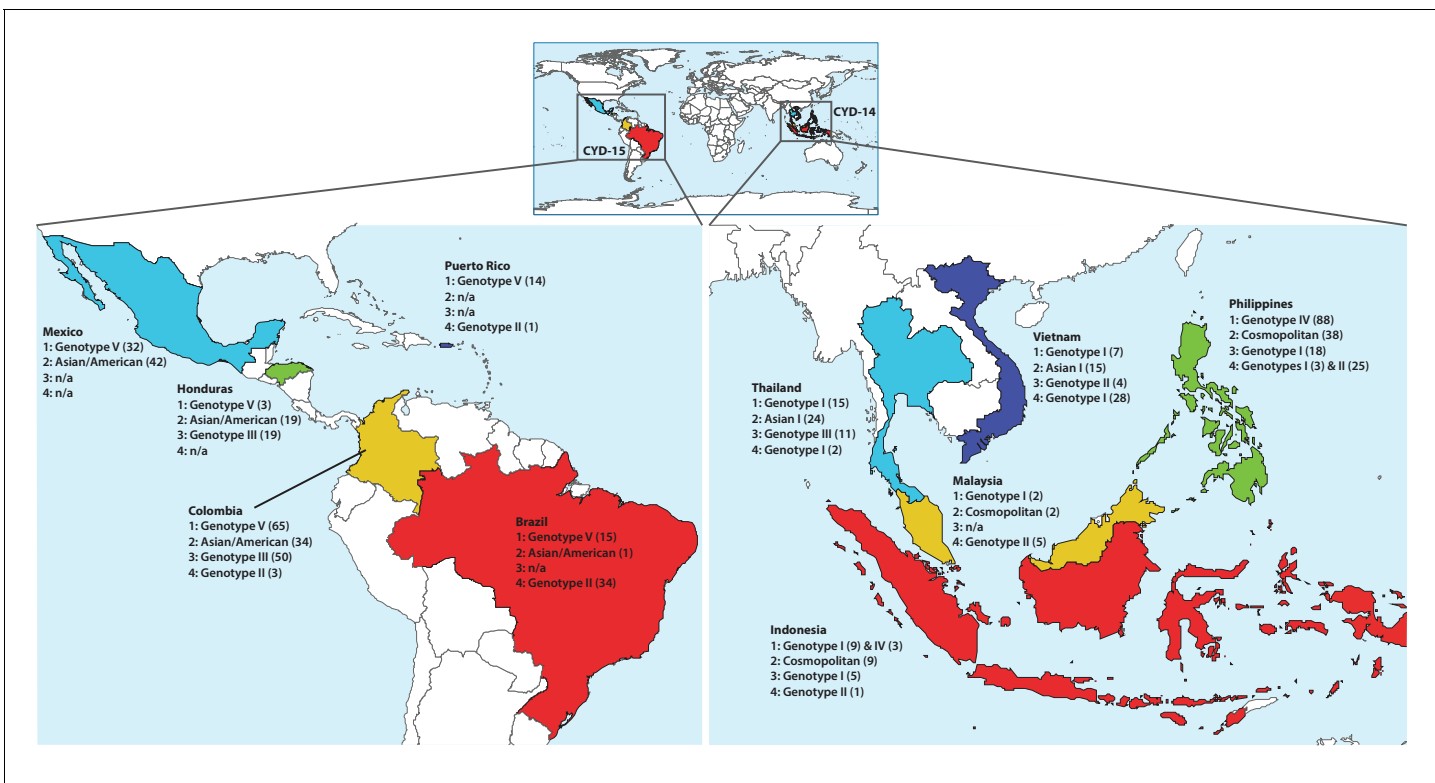

**Figure 2.** Distribution of DENV serotypes and genotypes sequenced in CYD14 and CYD15 vaccine trials by country. Numbers in parentheses indicate the total number of samples of each genotype for which complete or partial E gene sequences were obtained.

DOI: https://doi.org/10.7554/eLife.24196.009

The following figure supplements are available for figure 2:

**Figure supplement 1.** Genotype assignment of DENV-1 E gene sequences obtained from VCD samples in CYD14 and CYD15.

DOI: https://doi.org/10.7554/eLife.24196.010

**Figure supplement 2.** Genotype assignment of DENV-2 E gene sequences obtained from VCD samples in CYD14 and CYD15.

DOI: https://doi.org/10.7554/eLife.24196.011

**Figure supplement 3.** Genotype assignment of DENV-3 E gene sequences obtained from VCD samples in CYD14 and CYD15.

DOI: https://doi.org/10.7554/eLife.24196.012

**Figure supplement 4.** Genotype assignment of DENV-4 E gene sequences obtained from VCD samples in CYD14 and CYD15.

DOI: https://doi.org/10.7554/eLife.24196.013

## Human mAb epitope sequences in vaccine and wild-type viruses

We examined amino acid sequence identity between vaccine strains and wild-type CYD14/15 virus sequences at twelve B cell epitopes. The twelve epitopes represent some of the best structurally defined epitopes in DENV that are targeted by potent virus neutralising human mAbs and are thus of particular interest in vaccine development and immune correlate assays (*Fibriansah et al., 2014*; *Cockburn et al., 2012a*; *Smith et al., 2013*; *Fibriansah et al., 2015a, 2015b*; *Teoh et al., 2012*; *Rouvinski et al., 2015*; *Costin et al., 2013*; *Cockburn et al., 2012b*). Sequence analyses indicated limited variation at these epitope regions in the CYD14/15 sequences, as well as in a global database of wild-type virus sequences (*Figure 5* and *Figure 5—figure supplement 1*). The conservation of these epitope sequences between the decades-old 'donor' viruses from which the CYD-TDV product was derived and contemporary virus populations suggests that these amino acid sites are not highly prone to evolutionary drift.

## Vaccine efficacy by DENV serotype and genotype

Given the high degree of overall amino acid sequence identity, including at key epitope positions, between the E protein found in CYD-TDV vaccine strains and contemporary wild-type CYD14/15 viruses, we postulated that vaccine efficacy would be largely independent of virus genotype. We

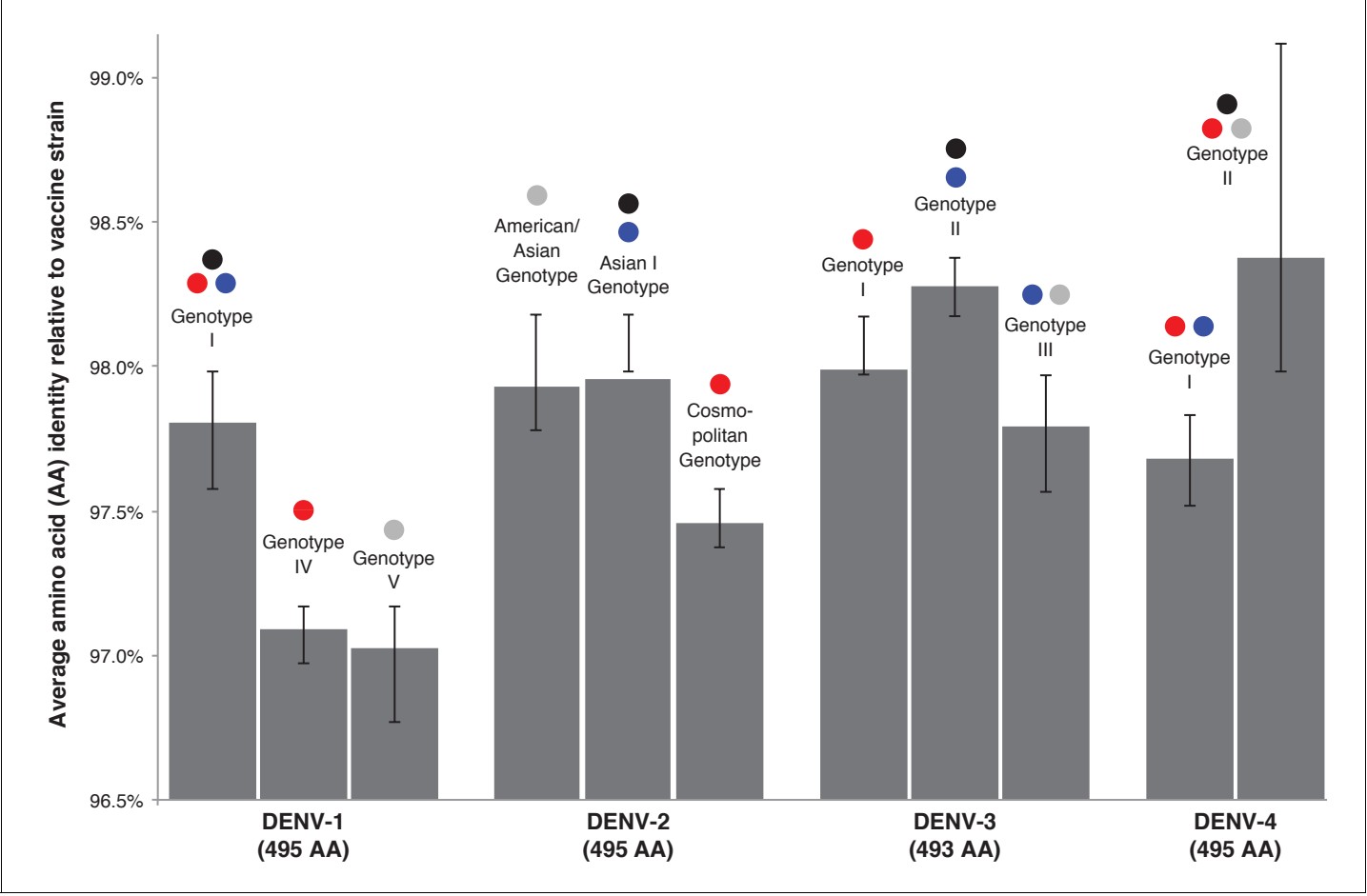

**Figure 3.** Average genotype-specific amino acid identity of DENV isolated in CYD-TDV trials compared to the vaccine strain of the corresponding DENV serotype. Black bars indicate the IQR of the full sample set. Coloured dots show the geographic regions from which each genotype was collected – red: CYD14, maritime SE Asia; blue: CYD14, mainland SE Asia; grey: CYD15, Americas. Black dots indicate the genotype of the serotype-specific CYD-TDV vaccine component.

DOI: https://doi.org/10.7554/eLife.24196.014

The following figure supplement is available for figure 3:

**Figure supplement 1.** Genotype-specific amino acid identity of DENV isolated in CYD-TDV trials compared to the vaccine strain of the corresponding DENV serotype versus vaccine efficacy.

DOI: https://doi.org/10.7554/eLife.24196.015

report two levels of intention to treat genotype-level efficacy from the CYD14 and CYD15 trials: the observed estimates and the observed+imputed estimates. The observed estimate refers to vaccine efficacy in the population of VCD cases who had serum samples yielding an E gene sequence that was empirically assigned a genotype. The observed+imputed estimates used the observed genotype data plus imputation to give genotype assignments to VCD cases where the serotype was known but genotype information was absent. Imputation was likely to be accurate because data from this study (*Figure 2*) indicated eight out of the ten study countries had only a single genotype of each serotype in circulation during the study period. Publicly available sequence data largely mirror the genotype distributions observed in this study; greater diversity is found in some Asian countries relative to those detected in this study, likely because the publicly available sequences are collated at the country level, whereas the CYD14/15 sequences represent those circulating only within the geographically limited trial populations (*Supplementary file 1c*). The count of observed and imputed genotypes is summarised in *Supplementary file 1d*.

Estimates of genotype-level vaccine efficacy amongst the observed and observed+imputed case populations are described in *Table 1* (all ages) and *Table 2* (participants 9–16 years of age). For

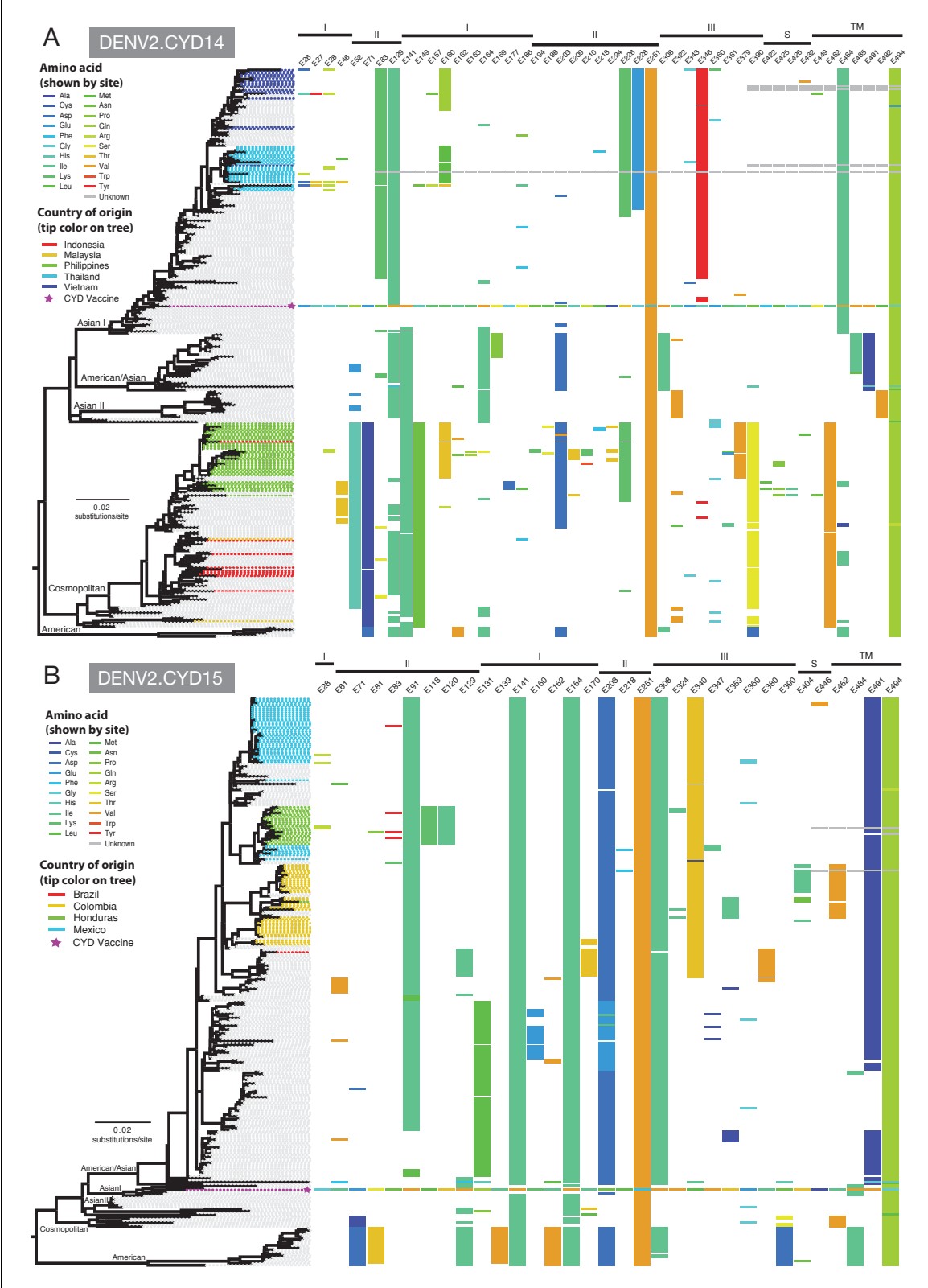

**Figure 4.** Amino acid differences between the DENV-2 E gene vaccine sequence, DENV-2 viruses isolated in CYD14 and CYD15 vaccine trials, and representative subsets of publically available DENV-2 sequences from the vaccine trial sites. (**A**) CYD14 DENV-2 phylogeny, (**B**) CYD15 DENV-2 phylogeny. Coloured tips on the trees show sequences isolated in the CYD-TDV trials (country of origin coloured as indicated in the key) and the vaccine sequence (purple star); grey tips indicate publically available sequences isolated from other studies in the countries of interest. Columns to the

*Figure 4 continued on next page*

*Figure 4 continued*

right indicate amino acid sites at which variation was observed in two or more CYD14/CYD15 sequences. Numbers at the top of columns indicate the amino acid site within the E gene. Bars at the top of the figures indicate the E gene domain of the site. Amino acids at variable sites in the E gene sequence of the vaccine component are shown in colour. For all other sequences, a lack of colour indicates an amino acid identical to that of the vaccine component at that site.

DOI: https://doi.org/10.7554/eLife.24196.016

The following figure supplements are available for figure 4:

**Figure supplement 1.** Amino acid differences between DENV-1 E gene vaccine sequences, DENV-1 viruses isolated in CYD14 vaccine trials, and representative subsets of publically available DENV-1 sequences from the vaccine trial sites.
DOI: https://doi.org/10.7554/eLife.24196.017

**Figure supplement 2.** Amino acid differences between DENV-1 E gene vaccine sequences, DENV-1 viruses isolated in CYD15 vaccine trials, and representative subsets of publically available DENV-1 sequences from the vaccine trial sites.
DOI: https://doi.org/10.7554/eLife.24196.018

**Figure supplement 3.** Amino acid differences between DENV-3 E gene vaccine sequences, DENV-3 viruses isolated in CYD14 vaccine trials, and representative subsets of publically available DENV-3 sequences from the vaccine trial sites.
DOI: https://doi.org/10.7554/eLife.24196.019

**Figure supplement 4.** Amino acid differences between DENV-3 E gene vaccine sequences, DENV-3 viruses isolated in CYD15 vaccine trials, and representative subsets of publically available DENV-3 sequences from the vaccine trial sites.
DOI: https://doi.org/10.7554/eLife.24196.020

**Figure supplement 5.** Amino acid differences between DENV-4 E gene vaccine sequences, DENV-4 viruses isolated in CYD14 vaccine trials, and representative subsets of publically available DENV-4 sequences from the vaccine trial sites.
DOI: https://doi.org/10.7554/eLife.24196.021

**Figure supplement 6.** Amino acid differences between DENV-4 E gene vaccine sequences, DENV-4 viruses isolated in CYD15 vaccine trials, and representative subsets of publically available DENV-4 sequences from the vaccine trial sites.
DOI: https://doi.org/10.7554/eLife.24196.022

completeness, we also show the observed genotype-level vaccine efficacy for participants < 9 years of age in *Supplementary file 2* but do not consider it in the main analyses because this age-class was only present in the CYD14 trial and is below the age for which the licensed vaccine is now indicated (i.e. $\geq$ 9 years) (*WHO, 2016*). For each serotype, a Cox proportional hazards regression model (expressing the hazard function) was used to estimate vaccine efficacy (derived as 100* [1- Hazard Ratio]) with vaccine group, genotype and the interaction between vaccine group and genotype included as covariates. The parameter estimates and the 95% confidence intervals of the interactions are given in *Table 3* (all ages) and *Table 4* (participants 9–16 years of age).

For DENV-1, vaccine efficacy estimates against the three different genotypes were highly similar in the all ages group and in participants 9–16 years of age (*Tables 1* and *2*). Additionally, the genotype interaction parameter estimates in the all ages group (*Table 3*) were close to zero and had reasonably tight 95% confidence bounds. This suggests it is unlikely that an interaction exists between genotype and vaccine efficacy, but if such an interaction does exist, it is small. Amongst participants 9–16 years of age, the interaction parameter estimates had 95% confidence intervals that bounded zero and were wider than the all ages group, making conclusions relatively difficult to draw.

For DENV-2, vaccine efficacy estimates against the American-Asian genotype (50.2%; 95% CI: 32.6–63.2%) and the Cosmopolitan genotype (43.8%; 95% CI: 16.1–62.2%) were similar, and both were higher than against the Asian I genotype (19.8%; 95% CI: −30.0–49.6%) in the all ages group (*Table 1*). However, the genotype interaction estimates had 95% confidence intervals that, although reasonably tight, included zero in the all ages group (*Table 3*) and in participants $\geq$ 9 years (*Table 4*). We note however that the upper bound of the confidence interval was very close to zero for the Asian/American versus Asian I genotype interaction (all ages, *Table 3*), leaving open the possibility that true heterogeneity may exist.

Within DENV-3, the confidence intervals for the interaction estimates were very wide when comparing genotype II versus genotype I in the all ages population (*Table 3*) and in participants $\geq$ 9 years (*Table 4*), and thus no conclusions could be drawn from these data. For DENV-3 genotype III versus genotype I, the 95% confidence intervals around the interaction estimates (*Tables 3* and *4*) were much tighter but nonetheless passed through zero. This suggested an interaction between genotype and vaccine efficacy remained possible but unlikely, and that more data would be needed to address this question.

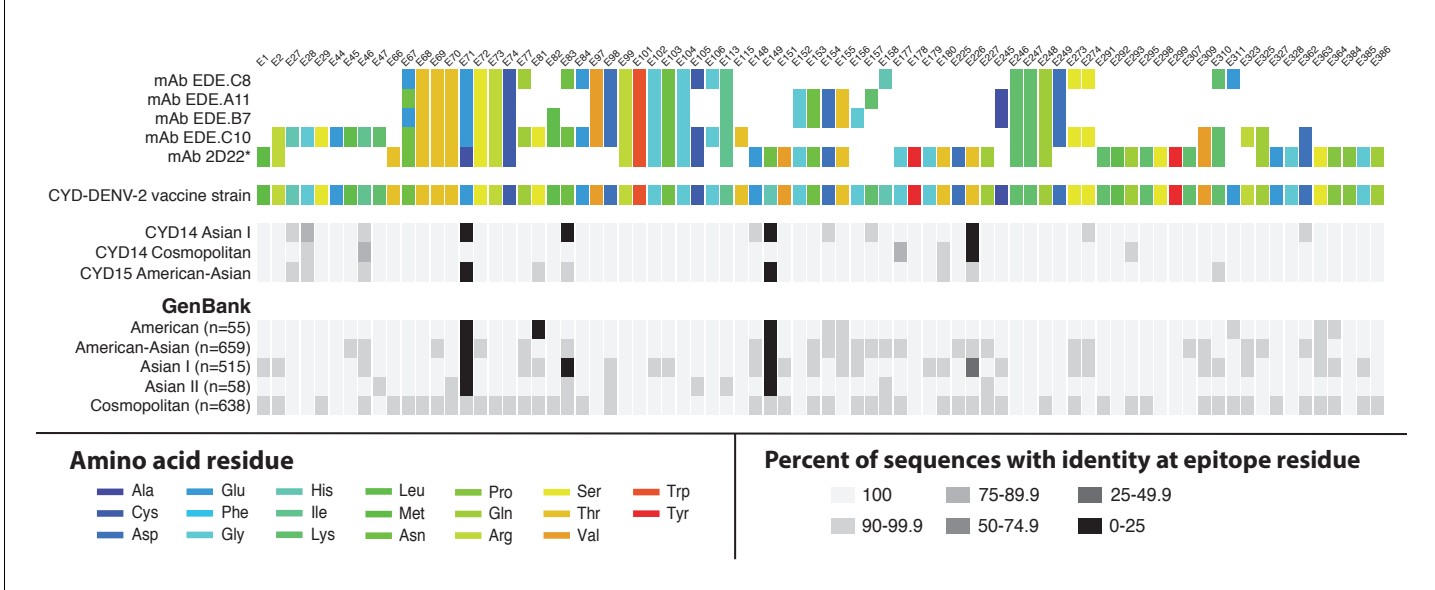

**Figure 5.** Sequence conservation between the DENV-2 vaccine component and wild-type DENV-2 viruses at epitope locations targeted by virus neutralising human mAbs. Amino acid targets for five neutralising human mAbs (*Fibriansah et al., 2015b*; *Rouvinski et al., 2015*) are coloured as indicated in the key (top) and compared to the vaccine sequence and wild-type sequences obtained within the CYD14 and CYD15 trials (middle), as well as complete E gene sequences from wild-type DENV-2 available on GenBank (bottom). Sites are indicated at the top of columns. For wild-type virus populations, the darker the block, the greater the proportion of sequences with an amino acid differing from the target amino acid at that site. When disagreement between amino acids was observed between epitope targets (as at E67 and E71), wild-type sequences were compared to 2D22 as a reference, denoted by an asterisk.

DOI: https://doi.org/10.7554/eLife.24196.023

The following figure supplement is available for figure 5:

**Figure supplement 1.** Sequence conservation between DENV vaccine components and wild-type DENV viruses at epitope locations targeted by virus neutralising human mAbs.

DOI: https://doi.org/10.7554/eLife.24196.024

Against DENV-4, vaccine efficacy was significantly lower against genotype I (58.3%, 95% CI: 29.9–75.2%), which circulates endemically only in Asia, compared to the globally distributed genotype II (81.8%, 95% CI: 74.3–87.1%, across CYD14/CYD15) in the all ages population (p=0.009) (*Table 1*). Confidence intervals around estimates of the interaction between genotype I and genotype II and vaccine group exclude zero, consistent with a lower efficacy against genotype I relative to genotype II (*Table 3*). However, when efficacy against DENV-4 genotype I versus genotype II was considered only in participants ≥ 9 years, efficacy was similar between genotypes (*Table 2*) and confidence intervals for interaction estimates included zero (*Table 4*). An important caveat is that relatively wide 95% confidence intervals for all interaction estimates amongst participants ≥ 9 years suggests limited power to detect differences in vaccine efficacy (*Table 4*), i.e. this study was generally underpowered to assess heterogeneity in this age subgroup analysis.

## Genotype-specific vaccine efficacy versus amino acid identity

A visualization of genotype-specific vaccine efficacy versus amino acid identity of trial viruses to CYD-TDV components is shown in *Figure 3—figure supplement 1*. These data illustrate the absence of a direct relationship between vaccine efficacy and genetic similarity between wild-type and vaccine strains of DENV.

## Discussion

The pivotal Phase III efficacy trials of CYD-TDV and longer term follow-up have revealed the complex efficacy profile of this vaccine (*Capeding et al., 2014*; *Hadinegoro et al., 2015*; *L'Azou et al., 2016*). These trials also highlighted generic challenges in dengue vaccine development, e.g. the

**Table 1.** Observed and imputed efficacy of CYD-TDV in all participants who received ≥1 injection (intention to treat) by serotype and genotype.

| | | CYD dengue vaccine group | | | Control group | | | Vaccine efficacy Observed | | Vaccine Efficacy with imputation for missing genotype data | |
|---|---|---|---|---|---|---|---|---|---|---|---|
| | | Cases | Person-years at risk | Density incidence (95% CI) | Cases | Person-years at risk | Density incidence (95% CI) | % | (95% CI) | % | (95% CI) |
| Serotype 1 | | | | | | | | 63.1 | (52.7; 71.2) | 54.7 | (45.4; 62.3) |
| | Genotype I CYD14[CYD] | 15 | 13742 | 0.1 (0.1; 0.2) | 18 | 6796 | 0.3 (0.2; 0.4) | 58.8 | (18.3; 79.5) | 57.4 | (29.7; 74.2) |
| | Genotype IV CYD14 | 40 | 13742 | 0.3 (0.2; 0.4) | 51 | 6796 | 0.8 (0.6; 1.0) | 61.3 | (41.5; 74.5) | 53.3 | (37.2; 65.3) |
| | Genotype V CYD15 | 53 | 27016 | 0.2 (0.1; 0.3) | 76 | 13434 | 0.6 (0.4; 0.7) | 65.3 | (50.9; 75.7) | 54.9 | (40.7; 65.6) |
| | p-value* | | | | | | | 0.8614 | | 0.9912 | |
| Serotype 2 | | | | | | | | 39.1 | (18.9; 54.3) | 43.0 | (29.4; 53.9) |
| | American/Asian CYD15 | 48 | 27035 | 0.2 (0.1; 0.2) | 50 | 13461 | 0.4 (0.3; 0.5) | 52.2 | (28.9; 67.9) | 50.2 | (32.6; 63.2) |
| | Asian I CYD14[CYD] | 28 | 13766 | 0.2 (0.1; 0.3) | 14 | 6856 | 0.2 (0.1; 0.3) | 0.3 | (−94.9; 46.6) | 19.8 | (−30.0; 49.6) |
| | Cosmopolitan CYD14 | 28 | 13766 | 0.2 (0.1; 0.3) | 21 | 6856 | 0.3 (0.2; 0.5) | 33.8 | (−18.0; 62.2) | 43.8 | (16.1; 62.2) |
| | p-value* | | | | | | | 0.1469 | | 0.2493 | |
| Serotype 3 | | | | | | | | 75.1 | (62.9; 83.3) | 71.6 | (63.0; 78.3) |
| | Genotype I CYD14 | 9 | 13835 | <0.1 (0.0; 0.1) | 14 | 6895 | 0.2 (0.1; 0.3) | 67.9 | (26.9; 86.6) | 58.1 | (25.2; 76.8) |
| | Genotype II CYD14[CYD] | 0 | 13835 | 0.0 (0.0; 0.0) | 4 | 6895 | <0.1 (0.0; 0.1) | 100.0 | (69.3; 100.0) | 85.8 | (41.1; 97.9) |
| | Genotype III CYD14 | 4 | 13835 | <0.1 (0.0; 0.1) | 7 | 6895 | 0.1 (0.0; 0.2) | 71.6 | (6.1; 92.6) | 68.4 | (19.8; 88.4) |
| | Genotype III CYD15 | 23 | 27060 | <0.1 (0.1; 0.1) | 47 | 13459 | 0.3 (0.3; 0.5) | 75.7 | (60.5; 85.5) | 74.2 | (64.3; 81.4) |
| | Genotype III CYD14 + CYD15 | 27 | 40896 | <0.1 (0.0; 0.1) | 54 | 20354 | 0.3 (0.2; 0.3) | 75.2 | (61.0; 84.6) | 73.7 | (64.3; 80.8) |
| | p-value* | | | | | | | 0.3751 | | 0.2561 | |
| Serotype 4 | | | | | | | | 74.1 | (61.7; 82.5) | 76.9 | (69.5; 82.6) |
| | Genotype I CYD14 | 19 | 13826 | 0.1 (0.1; 0.2) | 18 | 6874 | 0.3 (0.2; 0.4) | 47.4 | (−0.9; 72.5) | 58.3 | (29.9; 75.2) |
| | Genotype II CYD14[CYD] | 8 | 13826 | <0.1 (0.0; 0.1) | 24 | 6874 | 0.3 (0.2; 0.5) | 83.5 | (64.8; 93.1) | 83.8 | (69.3; 91.5) |
| | Genotype II CYD15[CYD] | 11 | 27063 | <0.1 (0.0; 0.1) | 31 | 13442 | 0.2 (0.2; 0.3) | 82.4 | (66.0; 91.5) | 80.8 | (71.2; 87.3) |
| | Genotype II CYD14 + CYD15[CYD] | 19 | 40890 | <0.1 (0.0; 0.1) | 55 | 20316 | 0.3 (0.2; 0.4) | 82.9 | (71.7; 90.1) | 81.8 | (74.3; 87.1) |
| | p-value* | | | | | | | 0.0072 | | 0.0086 | |

Cases: number of subjects with at least one sequenced symptomatic virologically-confirmed dengue episode during the active phase of follow-up.

Density incidence: data indicate cases per 100 person-years at risk.

*The p-value was obtained by testing the heterogeneity of genotype distribution between groups (within each serotype) using a Chi$^2$ (or Fisher's exact test).

[CYD] Genotype of the serotype-specific CYD-TDV vaccine component.

DOI: https://doi.org/10.7554/eLife.24196.025

**Table 2.** Observed and imputed efficacy of CYD-TDV for subjects 9 years and older who received ≥1 injection (intention to treat) by serotype and genotype

| | | CYD dengue vaccine group | | | Control group | | | Vaccine efficacy Observed | | Vaccine Efficacy with imputation for missing genotype data | |
|---|---|---|---|---|---|---|---|---|---|---|---|
| | | Cases | Person-years at risk | Density incidence (95% CI) | Cases | Person-years at risk | Density incidence (95% CI) | % | (95% CI) | % | (95% CI) |
| Serotype 1 | | | | | | | | 67.7 | (56.1; 76.3) | 58.4 | (47.7; 66.9) |
| | Genotype I CYD14[CYD] | 6 | 6683 | <0.1 (0.0; 0.2) | 8 | 3306 | 0.2 (0.1; 0.5) | 62.8 | (−6.8; 87.8) | 69.0 | (33.8; 85.5) |
| | Genotype IV CYD14 | 8 | 6683 | 0.1 (0.1; 0.2) | 19 | 3306 | 0.6 (0.3; 0.9) | 79.2 | (54.1; 91.4) | 64.0 | (39.7; 78.5) |
| | Genotype V CYD15 | 53 | 27016 | 0.2 (0.1; 0.3) | 76 | 13434 | 0.6 (0.4; 0.7) | 65.3 | (50.9; 75.7) | 54.9 | (40.7; 65.6) |
| | p-value* | | | | | | | 0.5213 | | 0.5400 | |
| Serotype 2 | | | | | | | | 48.6 | (27.4; 63.7) | 47.1 | (31.3; 59.2) |
| | American/Asian CYD15 | 48 | 27035 | 0.2 (0.1; 0.2) | 50 | 13461 | 0.4 (0.3; 0.5) | 52.2 | (28.9; 67.9) | 50.2 | (32.6; 63.2) |
| | Asian I CYD14[CYD] | 12 | 6687 | 0.2 (0.1; 0.3) | 9 | 3330 | 0.3 (0.1; 0.5) | 33.6 | (−62.7; 71.9) | 34.6 | (−27.4; 65.7) |
| | Cosmopolitan CYD14 | 5 | 6687 | <0.1 (0.0; 0.2) | 4 | 3330 | 0.1 (0.0; 0.3) | 37.8 | (−151; 83.5) | 40.3 | (−41.4; 74.3) |
| | p-value* | | | | | | | 0.7736 | | 0.7253 | |
| Serotype 3 | | | | | | | | 76.0 | (62.3; 84.7) | 73.6 | (64.4; 80.4) |
| | Genotype I CYD14 | 4 | 6715 | <0.1 (0.0; 0.2) | 6 | 3347 | 0.2 (0.1; 0.4) | 66.8 | (−16.3; 91.5) | 61.2 | (−4.1; 86.1) |
| | Genotype II CYD14[CYD] | 0 | 6715 | 0.0 (0.0; 0.1) | 3 | 3347 | <0.1 (0.0; 0.3) | 100.0 | (55.4; 100.0) | 80.1 | (7.6; 97.1) |
| | Genotype III CYD14 | 1 | 6715 | <0.1 (0.0; 0.1) | 2 | 3347 | <0.1 (0.0; 0.2) | 75.1 | (−160; 98.8) | 75.1 | (−27.4; 96.6) |
| | Genotype III CYD15 | 23 | 27060 | <0.1 (0.1; 0.1) | 47 | 13459 | 0.3 (0.3; 0.5) | 75.7 | (60.5; 85.5) | 74.2 | (64.3; 81.4) |
| | Genotype III CYD14 + CYD15 | 24 | 33775 | <0.1 (0.0; 0.1) | 49 | 16806 | 0.3 (0.2; 0.4) | 75.7 | (60.8; 85.3) | 74.3 | (64.7; 81.4) |
| | p-value* | | | | | | | 0.5928 | | 0.6985 | |
| Serotype 4 | | | | | | | | 85.2 | (74.6; 91.4) | 83.2 | (76.2; 88.2) |
| | Genotype I CYD14 | 3 | 6716 | <0.1 (0.0; 0.1) | 12 | 3327 | 0.4 (0.2; 0.6) | 87.6 | (60.9; 97.2) | 86.2 | (63.6; 94.8) |
| | Genotype II CYD14[CYD] | 3 | 6716 | <0.1 (0.0; 0.1) | 14 | 3327 | 0.4 (0.2; 0.7) | 89.4 | (67.7; 97.6) | 89.6 | (70.5; 96.3) |
| | Genotype II CYD15[CYD] | 11 | 27063 | <0.1 (0.0; 0.1) | 31 | 13442 | 0.2 (0.2; 0.3) | 82.4 | (66.0; 91.5) | 80.8 | (71.2; 87.3) |
| | Genotype II CYD14 + CYD15[CYD] | 14 | 33779 | <0.1 (0.0; 0.1) | 45 | 16769 | 0.3 (0.2; 0.4) | 84.6 | (72.6; 91.8) | 82.6 | (74.7; 88.1) |
| | p-value* | | | | | | | 1.0000 | | 0.6678 | |

Cases: number of subjects with at least one sequenced symptomatic virologically-confirmed dengue episode during the active phase of follow-up.

Density incidence: data indicate cases per 100 person-years at risk.

*The p-value was obtained by testing the heterogeneity of genotype distribution between groups (within each serotype) using a Chi$^2$ (or Fisher's exact test).

[CYD] Genotype of the serotype-specific CYD-TDV vaccine component.

DOI: https://doi.org/10.7554/eLife.24196.026

**Table 3.** Estimation of the interaction between genotype and vaccine group for symptomatic VCD detected during the active phase of follow-up by serotype in all participants who received >= 1 injection (intention to treat) (CYD14/CYD15).

The estimate of the interaction term between genotype and vaccine group is derived from Cox proportional hazards regression models including the vaccine group, the genotype and the interaction.

| Serotype | Parameter | Estimated interaction with observed vaccine efficacy | | Estimated interaction with vaccine efficacy with imputation | |
|---|---|---|---|---|---|
| | | Parameter estimate | 95% | Parameter estimate | 95% |
| Serotype 1 | Genotype IV vs Genotype I | −0.058 | [−0.858; 0.743] | 0.095 | [−0.475; 0.665] |
| | Genotype V vs Genotype I | −0.167 | [−0.936; 0.603] | 0.067 | [−0.492; 0.625] |
| Serotype 2 | American/Asian vs Asian I | −0.732 | [−1.486; 0.022] | −0.471 | [−1.032; 0.089] |
| | Cosmopolitan vs Asian I | −0.404 | [−1.259; 0.451] | −0.344 | [−0.966; 0.267] |
| Serotype 3 | Genotype II vs Genotype I | −12.748 | [−729.203; 703.707] | −1.079 | [−2.754; 0.596] |
| | Genotype III vs Genotype I | −0.251 | [−1.208; 0.705] | −0.459 | [−1.116; 0.198] |
| Serotype 4 | Genotype II vs Genotype I | −1.114 | [−1.943; −0.285] | −0.8184 | [−1.434; −0.203] |

DOI: https://doi.org/10.7554/eLife.24196.027

goal of balanced immunity to four DENV types, achieving efficacy in naïve and partially immune populations, and the need for long-term safety evaluation. A potential additional layer of complexity stems from the ongoing evolution of DENV populations in endemic countries and whether vaccines derived from viruses that circulated decades ago are 'fit for purpose' as immunogens against contemporary virus populations. Here, we demonstrate that the DENV E protein components in the CYD-TDV formulation shared high-level amino acid sequence identity, including at prominent B cell epitopes targeted by virus neutralising human mAbs, with viruses sampled in the CYD14 and CYD15 trials. Additionally, within the constraints of the available sample size and confounding factors discussed later, we found limited statistical evidence of genotype-specific differences in the efficacy profile.

With a total of 253 DENV-1, 191 DENV-2, 107 DENV-3 and 110 DENV-4 E gene sequences generated from CYD14 and CYD15, this study provides a contemporary characterization of DENV population genetics in ten highly endemic countries. Across all four serotypes, the E gene phylogenies positioned the CYD14 and CYD15 sequences together with those from geographically identical locations, consistent with long-term endemic circulation of a single virus genotype within each viral serotype in nearly all locations. Additionally, the phylogeographical profile demonstrates within-region sharing of virus genotypes but little inter-regional mixing (distinct viral population profiles can be distinguished within three major geographical categories: mainland Southeast Asia, maritime

**Table 4.** Estimation of the interaction between genotype and vaccine group for symptomatic VCD detected during the active phase of follow-up by serotype in subjects older than 9 years of age who received >= 1 injection (intention to treat) (CYD14/CYD15).

The estimate of the interaction term between genotype and vaccine group is derived from Cox proportional hazards regression models including the vaccine group, the genotype and the interaction.

| Serotype | Parameter | Estimated interaction with observed vaccine efficacy | | Estimated interaction with vaccine efficacy with imputation | |
|---|---|---|---|---|---|
| | | Parameter estimate | 95% | Parameter estimate | 95% |
| Serotype 1 | Genotype IV vs Genotype I | −0.574 | [−1.917; 0.768] | 0.153 | [−0.760; 1.066] |
| | Genotype V vs Genotype I | −0.061 | [−1.177; 1.054] | 0.385 | [−0.416; 1.186] |
| Serotype 2 | American/Asian vs Asian I | −0.327 | [−1.277; 0.624] | −0.270 | [−0.987; 0.448] |
| | Cosmopolitan vs Asian I | −0.064 | [−1.637; 1.510] | −0.089 | [−1.151; 0.972] |
| Serotype 3 | Genotype II vs Genotype I | −13.019 | [−943.634; 917.597] | −0.664 | [−2.578; 1.250] |
| | Genotype III vs Genotype I | −0.309 | [−1.665; 1.047] | −0.405 | [−1.443; 0.633] |
| Serotype 4 | Genotype II vs Genotype I | 0.226 | [−1.174; 1.626] | 0.244 | [−0.789; 1.277] |

DOI: https://doi.org/10.7554/eLife.24196.028

Southeast Asia, the Americas); only DENV-4 genotype II was found in multiple countries in both regions. The number of genotypes, and hence genetic diversity, detected for any given serotype was greater in Southeast Asia than in Latin America, as expected given the long history of hyperendemicity, within-serotype diversity, and high forces of infection in Southeast Asia (*Halstead, 2006*; *Holmes, 2009*; *Rodríguez-Barraquer et al., 2014*; *Imai et al., 2015*). Collectively, these data are informative for vaccine and drug development and design of molecular diagnostics. They also serve as virus population baseline profiles from which to monitor DENV evolution in countries where a selective pressure such as CYD-TDV might be widely introduced.

Indonesia, with two DENV-1 genotypes, and the Philippines, with two DENV-4 genotypes, were the only countries with genotype co-circulation detected within the trial. As each of these viral populations appears closely related to previously sequenced viruses from their respective country, these likely represent true local circulation in the population rather than recent importations of novel viruses. In general, however, long-term co-circulation of multiple genotypes within a single serotype in one location is rare and these viral populations may instead represent a cross-section of the viral populations during a process of genotype replacement, in which an endemic viral population is rapidly replaced (often completely) by a novel population imported from another geographical region (*Zhang et al., 2005*; *Lambrechts et al., 2012*; *Loroño-Pino et al., 2004*). This process may also be responsible for the presence of DENV-3 genotype III in Thailand, while neighboring Vietnam harbors only genotype II viruses. Genotype II was the dominant DENV-3 lineage in Thailand and mainland Southeast Asia from the early 1980s until at least 2010 (*Rabaa et al., 2013*), but was not detected in Thailand in this study. Recent reports of DENV-3 genotype III infections elsewhere in mainland Asia suggest that this lineage may be moving through the region, potentially replacing genotype II (*Lao et al., 2014*; *Jiang et al., 2012*). Ongoing virological surveillance in the CYD14 and CYD15 trial populations, as well as populations vaccinated post-licensure, will be used to further investigate the relationships between the vaccine, virus evolution and local DENV genotype variation.

Antigenic differences between viruses of the same serotype have been observed with neutralizing polyclonal and monoclonal antibodies in laboratory assays, and these differences have been postulated to be relevant to clinical epidemiology and vaccine development (*Katzelnick et al., 2015*; *Kochel et al., 2002*; *OhAinle et al., 2011*). Arguing against a critical role for within-serotype sequence diversity is the common acceptance that natural infection with one serotype elicits life-long clinical immunity to that serotype in the vast majority of instances (*Waggoner et al., 2016*). With respect to CYD-TDV, immunization of non-human primates elicited antibodies that neutralised geographically, phylogenetically and clinically diverse DENV serotypes and genotypes (*Barban et al., 2012*). That CYD-TDV immunisation induced similar measured levels of efficacy against all genotypes of DENV-1 is supportive of the concept that clinical immunity elicited by CYD-TDV to this serotype was pan-genotype in nature (*Tables 1* and *2*). Analyses also suggest potential pan-genotype immunity in the case of DENV-3, although the relatively low numbers of DENV-3 cases detected within the CYD14 trials in Southeast Asia result in this study being underpowered to assess potential heterogeneity. Further research is warranted to understand if smaller differences exist in genotype-level vaccine efficacy than could be measured in these trials and that might be relevant to programmatic use of vaccine.

In DENV-4, efficacy against genotype I (found in Asia) in the all ages population was significantly lower than that against genotype II in both Asia and the Americas. Subgroup analyses of genotype and age-stratified vaccine efficacy are inevitably speculative because of diminishing sample sizes and wide confidence intervals around the interaction estimates. Nonetheless we observed that in participants 9–16 years of age, the age group eligible for the licensed vaccine, the vaccine efficacy point estimates were similarly high (>80%) between DENV-4 genotypes.

DENV-2 is of particular interest because CYD-TDV efficacy is lowest against this serotype. In a previous, single-centre phase IIb trial (CYD23) in Thai children (*Sabchareon et al., 2012*), where all the circulating DENV-2 viruses belonged to the Asian I genotype, efficacy was just 3.5% (-59.8; 40.5) against this serotype/genotype. In CYD14/15, with a larger sample size, efficacy against the DENV-2 Asian I genotype in the all ages population (19.8% (-30.0; 49.6)) was lower, but not significantly so, than that against other DENV-2 genotypes. Although heterogeneity could not be confirmed, further analysis of the interaction between genotype and vaccine group suggested a potentially decreased efficacy profile of the DENV-2 Asian I genotype in the all ages population compared to the American/Asian genotype, which currently circulates only in the Americas. As above for DENV-4, it is

speculative to examine subgroups, but for the age group 9–16 years of age, the age group eligible for the now licensed vaccine, the efficacy against DENV-2 Asian I genotype (34.6% (-27.4; 65.7) was comparable to that seen against the other DENV-2 genotypes (Asian/American and Cosmopolitan), albeit with inevitably wide confidence intervals around the point estimates. The basis for reduced efficacy against DENV-2 Asian I genotype and DENV-4 genotype I in the all ages population could be complex and linked to uncharacterised differences in how vaccine-elicited immunity acts on these virus genotypes. We note that DENV-2 Asian I genotype and DENV-4 genotype I viruses were only detected in Asia (CYD14), where younger trial participants were included compared to Latin America (CYD15). While a sole impact of age in vaccine-elicited immunity may account for a proportion of this difference, high DENV diversity within Asia resulted in the detection of additional DENV-2 and DENV-4 genotypes within the CYD14 study (DENV-2 Cosmopolitan genotype and DENV-4 genotype II), against which no evidence of decreased vaccine efficacy was shown. It will be of interest to monitor efficacy against DENV-2 and DENV-4 genotypes in post-marketing effectiveness studies of CYD-TDV. Interestingly, while the parental strain of the CYD DENV-2 component is in fact based on an historical Asian I strain, contemporary DENV-2 Asian I populations diverge from the parental CYD strain at multiple amino acid residues, some of which are postulated to increase transmission fitness in some circumstances (*Vu et al., 2010*). Additional investigations, which might include animal model studies coupled with virological surveillance in the post-vaccine licensure period, could assist further understanding and precision of estimates of vaccine efficacy against different genotypes of DENV-2.

Examination of amino acid sequences among circulating viruses, vaccine components, and epitope sequences targeted by potent, virus neutralising human mAbs provides a framework to predict and possibly understand genotype-specific vaccine performance. These analyses generally underscore the similarity between vaccine components and circulating viruses. Where there were differences at epitope sequence locations, we did not observe a measurable effect in the genotype-level vaccine efficacy. For example, mismatches between circulating DENV-1 and the vaccine component at IF4 epitope sites (*Fibriansah et al., 2014*) (sites E155, E161, and E171; *Figure 5—figure supplement 1*) are present across individual genotypes, yet the point estimates of genotype-specific vaccine efficacy were not measurably different. In DENV-4, there was evidence of lower vaccine efficacy against genotype I viruses and also amino acid mismatches between the vaccine component and genotype I virus sequences at known 5H2 epitope positions (E155 and E160; *Figure 5—figure supplement 1*) (*Cockburn et al., 2012a*). Further research will be needed to understand the significance of these differences for clinical immunity.

Several mismatches between circulating DENV-2 viruses and the vaccine component were observed at important epitope sites (*Figure 5*), but these were shared by two circulating genotypes in most cases (sites E71, E149 and E226). Comparing wild-type DENV-2 sequences sampled in CYD14/CYD15 to the CYD-TDV vaccine component, only site E83 showed a mismatch in a high proportion of the Asian I population alone. Sequence data obtained from GenBank confirm that, while there is some variability at this site among all contemporaneous DENV-2 lineages, the Asian I lineage is defined by this amino acid difference at E83. An important caveat to these sequence comparisons is that amino acid differences between vaccine strains and wild-type viruses at known B cell epitopes does not necessarily imply that antigenicity (or immunogenicity) is altered – functional assays of antibody binding will be needed for this.

Our study had several limitations. These were *post hoc* analyses and, inevitably, sample sizes became small for genotype-level vaccine efficacy estimates and in particular the age-class (9–16 year olds) subgroup analysis. This manifests as wide 95% confidence intervals around the genotype-level point estimates of vaccine efficacy. We generally did not obtain E gene sequences from VCD cases with low viremia and, hence, whether some rare genotypes are not represented in the population of E gene sequences is unknown. In countries where only a single genotype was detected, it was assumed that no undetected genotypes were circulating concurrently in the same location, while publicly available data indicate greater diversity in some Asian countries during this period than detected in this study (Serotypes 1 and 3; *Supplementary file 1c*). This assumption may have affected the imputed estimates of vaccine efficacy within CYD14, but because vaccine efficacy was ultimately estimated across the entire study population rather than at the country level, the impact of this assumption is expected to be limited. Baseline serostatus is an important determinant of the efficacy profile of CYD-TDV (*Hadinegoro et al., 2015*). It is possible that some of the genotype-level efficacy results are confounded by the baseline serostatus of vaccine recipients in particular

countries, but because only 10% of all participants were characterized immunologically at baseline it is not possible to explore this further (*Capeding et al., 2014*). Such questions may be addressed in post-licensure research. Finally, our results and conclusions may be specific to this particular vaccine given its unique composition. Other vaccines in development employ different donor viruses and have different compositions and, hence, deserve their own evaluations; in any large-scale trial of a candidate DENV vaccine, continual monitoring will be key to understanding the landscape and evolution of circulating DENV populations and further elucidating the potential relationships between virological factors, vaccine efficacy and post-immunization transmission dynamics. Despite these limitations, the results described here improve the understanding of CYD-TDV vaccine performance. Post-licensure research is needed to further understand the complex profile of this vaccine, and to monitor the impact of vaccination programs on the evolution of DENV populations.

## Materials and methods

### Sample sets

Briefly, viremic serum samples from VCD cases detected during the active phase of surveillance in CYD14 (Indonesia, Malaysia, the Philippines, Thailand, Vietnam; collected between June 2011 and December 2013) and CYD15 (Brazil, Colombia, Honduras, Mexico, Puerto Rico; collected between June 2011 and April 2014) were eligible for inclusion in this study (*Figure 1*). Samples with very low viremia or low sample volume that were highly unlikely to be fit for purpose with respect to nucleic acid isolation, amplification and sequencing of prM/E genes were excluded from further investigation, as were samples for which consent was not obtained.

### Sequencing methodology

A total of 433 and 512 viremic serum samples were available and subjected to sequencing from CYD14 and CYD15, respectively. Viral RNA was extracted from serum using the MagNA Pure 96 DNA and Viral NA Small Volume Kit (Roche, Mannheim, Germany). PrM and E gene regions were amplified by PCR using 16 different primer pairs, with universal tails at the 5′ end to allow the addition of 454 sequencing-specific nucleotides and isolate-specific multiplex identifiers (MIDs) in a second PCR round, 'barcode incorporation PCR'. The first PCR round was performed in 20 µl reaction volumes using the FastStart High Fidelity Reaction Kit (Roche) with the addition of 0.25 µM of each PCR primer. The target genes were amplified by PCR in 96 well plates, with the following cycling conditions: denaturation at 94°C for 2 min followed by 40 cycles of PCR, with cycling conditions of 30 s at 94°C, 1 min at 57°C for DENV-1; 60°C for DENV-2; 56°C for DENV-3; 55°C for DENV-4, 60 s at 72°C and 72°C in 5 min for final extension. After PCR, the amplicons were purified using magnetic AMPure XP beads (Agencourt, Woerden, The Netherlands).

### Barcode incorporation PCR

The purified first round PCR amplicons were re-amplified to incorporate 454 sequencing-specific nucleotides and isolate-specific MIDs. For this, we used fusion primers that are composed of three parts: 454 sequencing-specific adapter nucleotides, MID sequences and the sequence target of interest on the DNA sample. The second PCR reactions were performed in 10 µl reaction volumes using the FastStart High Fidelity Reaction Kit (Roche) with the addition of 0.1 µM of each fusion primer. The thermo cycling conditions were: denaturation at 94°C for 2 min followed by 30 cycles of PCR, with cycling conditions of 30 s at 94°C, 30 s at 57°C, 60 s at 72°C and 72°C in 5 min for final extension.

### Sample pooling

Amplicons spanning the same genomic coordinates, but from different viruses, were pooled. Amplicon pools were measured using Quant-iT PicoGreen dsDNA Assay Kit (Invitrogen, Carlsbad, California) after purification by magnetic AMPure XP beads. In preparation for 454 sequencing, the concentration of the pooled amplicons was adjusted to $10^6$ copies/mL. The purified amplicons were the pooled into one library tube at a concentration of $5.10^5$ copies/mL.

## Emulsion PCR and 454 sequencing

An emulsion-based clonal amplification (emPCR) was performed according to the manufacturer's instructions as described in the emPCR Amplification Method Manual - Lib-A, revision June 2010 (Roche). DNA sequencing was performed using the GS Junior Titanium Sequencing Kit and the GS Junior Titanium PicoTiterPlate using the Sequencing Method Manual, revision June 2010 (Roche).

## Data analysis

GS Mapping software (Roche) was used for primer trimming and alignment of reads against reference sequences. Briefly, each read per amplicon was mapped to a reference sequence (DENV-1/VN/BID-V2732/2007, GenBank accession number GQ199773.1; DENV-2/VN/BID-V1873/2007, GenBank accession number FJ461321.1; DENV-3/VN/BID-V1933/2008, GenBank accession number KF955460.1; DENV-4/KH/BID-V2055/2002 (GenBank accession number KF955510.1). Sequence quality was high; the Phred scores for E gene sequences are provided in *Supplementary file 3*. Technical controls were included in all sequencing runs, and showed 100% sequence concordance across the prM and E gene in all cases. CYD-TDV prME sequences from CYD1-CYD4 vaccine components are deposited in GenBank under accession numbers KX239894-KX239897, respectively. All sequences obtained from study subjects are deposited in GenBank under accession numbers KY818060-KY818289, KY851378-KY851758, and KY882502-KY882554.

## Sequence analysis

A total of 664 DENV prM and/or E gene sequences were obtained using the above protocol (DENV-1, 253; DENV-2, 191; DENV-3, 108; DENV-4, 112) (*Figure 1*, *Figure 1—source datas 1–4*). Sequences were manually aligned using Geneious (v7.1.7; RRID:SCR_010519) and validated on both nucleotide and amino acid levels. Due to the availability of a larger, less geographically biased public database of E gene sequences compared to prM/E, we focused analysis on the E gene only (1485 nucleotide/495 amino acids for DENV-1,–2, −4; 1479 nucleotide/493 amino acids for DENV-3), excluding three sequences for which only the prM sequence could be obtained. Thus for each serotype, all full and partial E gene sequences (DENV-1, 253; DENV-2, 191; DENV-3, 107; DENV-4, 110) were aligned to large datasets of publicly available E gene sequences from GenBank for which the country and year of sampling are known. Maximum likelihood (ML) phylogenies were inferred for nucleotide sequences using RAxML (v8.0, http://www.exelixis-lab.org/; RRID:SCR_006086) under the GTRGAMMAI model and were visually assessed to determine the genotype of each virus obtained from CYD14 and CYD15. Visual inspection further indicated that all viruses sequenced in this study fell into expected lineages corresponding to previously sampled sequences from the countries from which they were isolated.

To investigate DENV sequences from the CYD14 and CYD15 studies in the context of the viruses circulating in their respective countries and make datasets more tractable, datasets were sub-sampled to include all CYD14 and CYD15 E gene sequences and up to three randomly selected, publicly available sequences per country per year from the countries involved in this study, along with up to five representative sequences of each known genotype (*Goncalvez et al., 2002*; *Twiddy et al., 2002*; *Wittke et al., 2002*; *Patil et al., 2012*), regardless of the country from which they were isolated (Total number of taxa used for phylogenetic reconstruction, CYD14: DENV-1, 317; DENV-2, 279; DENV-3, 222; DENV-4, 207. Total number of taxa used for phylogenetic reconstruction, CYD15: DENV-1, 236; DENV-2, 252; DENV-3, 159; DENV-4, 119). Evolutionary models for each dataset were determined using jModeltest (v2.0; RRID:SCR_015244) (*Posada, 2009*). ML trees were then inferred from these nucleotide sequences using RAxML (v8.0) under the GTRGAMMAI model with 500 bootstrap replications, and *p*-uncorrected sequence identity (pairwise comparison of genetic differences across the nucleotide and amino acid alignments) was determined using Geneious (v7.1.7). To investigate potentially novel amino acid residues or changes suggestive of selection, all amino acid sites showing a difference between two or more CYD14/CYD15 sequences relative to the vaccine or circulating viruses of the same genotype were mapped to the aforementioned phylogenies using Phandango (https://jameshadfield.github.io/phandango; RRID:SCR_015243). All phylogenies were visualized and annotated using FigTree (v1.4.2; RRID:SCR_008515) and Phandango. Gene annotations were done using the GR7 sequence viewer.

## Epitope mapping

To assess the diversity of DENV amino acid sequences and vaccine strains at sites targeted by virus neutralising human mAbs, the serotype-specific E gene alignments used for genotype determination were trimmed to include only sites at which a relevant epitope has previously been identified (*Fibriansah et al., 2014*; *Cockburn et al., 2012a*; *Smith et al., 2013*; *Fibriansah et al., 2015a*; *2015b*; *Teoh et al., 2012*; *Rouvinski et al., 2015*; *Costin et al., 2013*; *Cockburn et al., 2012b*). For each site, the sequences were compared to vaccine components, strains isolated in CYD14 and CYD15, and publicly available sequences to determine the frequency at which viral sequences matched mAbs targets across all known human DENV lineages.

## Statistical determination of genotype-specific vaccine efficacy

Vaccine efficacy against symptomatic VCD cases according to each genotype during the active phase (i.e. from D0 to Month 25) was calculated using the number of cases (i.e., children/adolescents with one or more episodes of VCD) and the person-time at risk in all participants who received at least one injection according to intention to treat. The incidence density was derived as the number of cases per 100 person-years at risk in each group. A Cox regression model was used to estimate vaccine efficacy (derived as 100* [1- Hazard Ratio]) with vaccine group included as a covariate and 95% CI. To further investigate the interaction between vaccine efficacy and genotype, an additional Cox proportional hazards regression model (expressing the hazard function) was used to estimate vaccine efficacy with vaccine group, genotype and the interaction between vaccine group and genotype included as covariates.

VCD cases with missing genotype were imputed using multiple imputation techniques (logistic regression) by serotype with the country included in the model. The twenty imputed (completed) datasets were then analyzed separately and the resultant estimates combined using Rubin's variance rules and their multivariate generalizations (*Rubin, 1987*). Analyses were run based on the available data (i.e. no imputed values) and on the imputed data (raw imputation data are shown in *Supplementary file 1d*).

A Chi$^2$ test (or Fisher's exact test) was used to test the heterogeneity of genotype distribution between vaccine groups. The alternative procedure for pooling chi-square distributed statistics that was proposed by Rubin (*Rubin, 1987*) and further investigated by Li *et al.* was used on imputed data (*Li et al., 1991*). A p-value of less than 0.10 was considered to indicate statistical significance. All statistical analyses were performed using SAS (Version 9.3; RRID:SCR_008567).

## Acknowledgements

The authors would like to acknowledge all investigators, Sanofi Pasteur's Global Clinical Immunology team and volunteers involved in the clinical evaluation of CYD-TDV. Many thanks also to all Sanofi Pasteur Dengue Vaccine team members and in particular to Helena Aurell, Alain Bouckenooghe, Laurent Chambonneau, Danaya Chansinghakul, Diana Leticia Coronel, Remi Forrat, Carina Frago, Etienne Gransard, Thelma Laot, Josemund Menezes, Fernando Noriega, Leon Ochiai, Eric Plennevaux, Paula Perroud, Enrique Rivas, Melanie Saville, Jo-Ann West and Jean Lang for their support. In Vietnam, thanks to Vi Tran Thuy and Huy Huynh Le Anh. We would also like to thank Hao Chung The and Christine Boinett for assistance with figures. A special thanks to Bruno Guy for helpful discussions and critical reading of the manuscript.

## Additional information

### Competing interests

Yves Girerd-Chambaz, Matthew Bonaparte, Diane van der Vliet, Edith Langevin, Margarita Cortes, Betzana Zambrano, Corinne Dunod, Anh Wartel-Tram, Nicholas Jackson: Employee of Sanofi Pasteur, a company engaged in the development of CYD-TDV. The other authors declare that no competing interests exist.

## Funding

| Funder | Grant reference number | Author |
| --- | --- | --- |
| Sanofi Pasteur | Direct funding to perform viral sequencing | Cameron P Simmons |

The funders had no role in study design, data collection and interpretation, or the decision to submit the work for publication.

## Author contributions

Maia A Rabaa, Conceptualization, Resources, Data curation, Software, Formal analysis, Supervision, Funding acquisition, Validation, Investigation, Visualization, Methodology, Writing—original draft, Writing—review and editing; Yves Girerd-Chambaz, Conceptualization, Resources, Data curation, Software, Formal analysis, Supervision, Funding acquisition, Validation, Investigation, Visualization, Methodology, Writing—original draft, Project administration, Writing—review and editing; Kien Duong Thi Hue, Investigation, Visualization, Methodology, Writing—original draft, Writing—review and editing; Trung Vu Tuan, Investigation, Methodology, Writing—review and editing; Bridget Wills, Supervision, Writing—review and editing; Matthew Bonaparte, Diane van der Vliet, Edith Langevin, Margarita Cortes, Betzana Zambrano, Corinne Dunod, Anh Wartel-Tram, Data curation, Writing—review and editing; Nicholas Jackson, Conceptualization, Data curation, Funding acquisition, Project administration, Writing—review and editing; Cameron P Simmons, Conceptualization, Resources, Supervision, Funding acquisition, Investigation, Visualization, Writing—original draft, Project administration, Writing—review and editing

## Author ORCIDs

Maia A Rabaa, http://orcid.org/0000-0003-0529-2228
Yves Girerd-Chambaz, http://orcid.org/0000-0002-5121-686X
Cameron P Simmons, http://orcid.org/0000-0002-9039-7392

## Ethics

Clinical trial registration ClinicalTrials.gov Identifiers: NCT01373281 & NCT01374516
The CYD14 and CYD15 studies were conducted in compliance with Good Clinical Practice guidelines, the principles of the Declaration of Helsinki, and the regulations of the relevant countries. Each study was approved by the appropriate ethics review committee and regulatory agencies as per local rules. Written informed consent was obtained from a parent or guardian for all participants in the two trials, with assent obtained depending on the age of the participant. Metadata for viruses sequenced in this study included the date and country of isolation; to maintain anonymity, no other defining characteristics of study participants were attached to viral data.

## Decision letter and Author response

Decision letter https://doi.org/10.7554/eLife.24196.055
Author response https://doi.org/10.7554/eLife.24196.056

# Additional files

## Supplementary files

• Supplementary file 1 . (a) Sequencing success rates in samples from virologically confirmed dengue cases in vaccine and control groups from the CYD-TDV trials. (b) Mean percent identity between E gene amino acid sequences of the relevant serotype-specific CYD-TDV vaccine strain and virus populations sampled in CYD14/15. (c) Number of E gene sequences per genotype per country in CYD-TDV trials versus publicly available sequences on GenBank. I. CYD14, II. CYD15. (1d) Variation in the number of cases, imputed versus observed.
DOI: https://doi.org/10.7554/eLife.24196.029

• Supplementary file 2 . Observed and imputed efficacy of CYD-TDV for subjects less than 9 years of age who received ≥1 injection (intention to treat) by serotype and genotype.

DOI: https://doi.org/10.7554/eLife.24196.030

• Supplementary file 3. Phred scores indicating sequence quality for all CYD14/15 DENV prM/E sequences.
DOI: https://doi.org/10.7554/eLife.24196.031
• Transparent reporting form
DOI: https://doi.org/10.7554/eLife.24196.032

## Major datasets

The following datasets were generated:

| Author(s) | Year | Dataset title | Dataset URL | Database, license, and accessibility information |
|---|---|---|---|---|
| Rabaa MA, Chambaz YG, Duong KTH, Vu TT, Wills B, Bonaparte M, Vliet DVD, Langevin E, Cortes M, Zambrano B, Dunod C, Tram AW, Jackson N, Simmons CP | 2017 | Dengue virus prM/E genes from CYD-TDV trials | https://www.ncbi.nlm.nih.gov/nuccore/?term=KY818060%3AKY818289%5Baccn%5D | Publicly available at NCBI Nucleotide (accession no: KY818060-KY818289) |
| Rabaa MA, Chambaz YG, Duong KTH, Vu TT, Wills B, Bonaparte M, Vliet DVD, Langevin E, Cortes M, Zambrano B, Dunod C, Tram AW, Jackson N, Simmons CP | 2017 | Dengue virus prM/E genes from CYD-TDV trials | https://www.ncbi.nlm.nih.gov/nuccore/?term=KY882502%3AKY882554%5Baccn%5D | Publicly available at NCBI Nucleotide (accession no: KY882502-KY882554) |
| Rabaa MA, Chambaz YG, Duong KTH, Vu TT, Wills B, Bonaparte M, Vliet DVD, Langevin E, Cortes M, Zambrano B, Dunod C, Tram AW, Jackson N, Simmons CP | 2017 | Dengue virus prM/E genes from CYD-TDV trials | https://www.ncbi.nlm.nih.gov/nuccore/?term=KY851378%3AKY851758%5Baccn%5D | Publicly available at NCBI Nucleotide (accession no: KY851378-KY851758) |

The following previously published datasets were used:

| Author(s) | Year | Dataset title | Dataset URL | Database, license, and accessibility information |
|---|---|---|---|---|
| Mantel N, Girerd Y, Geny C, Bernard I, Pontvianne J, Lang J, Barban V | 2016 | Synthetic construct isolate CYD1 surface protein gene, partial cds | https://www.ncbi.nlm.nih.gov/nuccore/KX239894 | Publicly available at NCBI Nucleotide (accession no: KX239894) |
| Mantel N, Girerd Y, Geny C, Bernard I, Pontvianne J, Lang J, Barban V | 2016 | Synthetic construct isolate CYD2 surface protein gene, partial cds | https://www.ncbi.nlm.nih.gov/nuccore/KX239895 | Publicly available at NCBI Nucleotide (accession no: KX239895) |
| Mantel N, Girerd Y, Geny C, Bernard I, Pontvianne J, Lang J, Barban V | 2016 | Synthetic construct isolate CYD3 surface protein gene, partial cds | https://www.ncbi.nlm.nih.gov/nuccore/KX239896 | Publicly available at NCBI Nucleotide (accession no: KX239896) |
| Mantel N, Girerd Y, Geny C, Bernard I, Pontvianne J, Lang J, Barban V | 2016 | Synthetic construct isolate CYD4 surface protein gene, partial cds | https://www.ncbi.nlm.nih.gov/nuccore/KX239897 | Publicly available at NCBI Nucleotide (accession no: KX239897) |

| Henn MR, Young S, Koehrsen M, Lennon N, Erlich R, et al | 2009 | Dengue virus 1 isolate DENV-1/VN/BID-V2732/2007, complete genome | https://www.ncbi.nlm.nih.gov/nuccore/GQ199773.1 | Publicly available at NCBI Nucleotide (accession no: GQ199773) |
| Henn MR, Young S, Koehrsen M, Lennon N, Erlich R, et al | 2009 | Dengue virus 2 isolate DENV-2/VN/BID-V1873/2007, complete genome | https://www.ncbi.nlm.nih.gov/nuccore/FJ461321.1 | Publicly available at NCBI Nucleotide (accession no: FJ461321) |
| Henn MR, Young S, Koehrsen M, Lennon N, Erlich R, et al | 2014 | Dengue virus 3 isolate DENV-3/VN/BID-V1933/2008, complete genome | https://www.ncbi.nlm.nih.gov/nuccore/KF955460.1 | Publicly available at NCBI Nucleotide (accession no: KF955460) |
| Henn MR, Young S, Koehrsen M, Lennon N, Erlich R, et al | 2014 | Dengue virus 4 isolate DENV-4/KH/BID-V2055/2002, complete genome | https://www.ncbi.nlm.nih.gov/nuccore/KF955510.1 | Publicly available at NCBI Nucleotide (accession no: KF955510) |

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
