## [Decision Letter]

Thank you for submitting your article "Genetic epidemiology of dengue virus in phase III trials of the CYD tetravalent vaccine and implications for efficacy" for consideration by *eLife*. Your article has been reviewed by three peer reviewers, and the evaluation has been overseen by a Reviewing Editor and Wendy Garrett as the Senior Editor. The reviewers have opted to remain anonymous.

The reviewers have discussed the reviews with one another and the Reviewing Editor has drafted this decision to help you prepare a revised submission.

Summary:

This further analysis of two dengue vaccine trials reports sequencing and phylogenetic results for viruses that were isolated from vaccinated and control cases in the trials, examines the protein location of the variation along the alignment and with respect to antibody epitopes, and concludes that serotype-specific vaccine efficacy is relatively homogeneous across the genetic variation within each serotype (a conclusion challenged by the reviewers). It provides very interesting data on the relationship of vaccine viruses to circulating viruses and some information on the relationship between this and vaccine efficacy, but important issues remain with the analysis and interpretation.

Essential revisions:

It is *eLife*'s policy to provide one list of changes requested for reconsideration of a revised manuscript. We do that in what follows, and would like point by point responses to the overall decision. Because there is some extra detail in the individual reviewer comments, I have chosen to include them in their entirety for your benefit. I have lifted language from them to inform the main decision comments, so you will see some redundancy. You do not need to write point-by-point responses to each reviewer’s comments.

Overall decision:

We invite a revised resubmission that addresses the following essential points (A-D):

A) The most important theme in the comments from all reviewers is, in brief, that the authors have interpreted an absence of evidence to reject the null hypothesis of homogeneous efficacy against each serotype (across genotypes), as evidence in favor of the null. Statistically, this is invalid, and from a scientific and practical perspective it seems to give an unduly rosy picture of the efficacy of the vaccine and unduly low weight to concerns about evolution to escape the protection of the vaccine. As reviewer 1 writes: "In this manuscript authors Rabaa and colleagues investigate the important issue of whether the CYD14 and CYD15 phase III dengue vaccine trials showed differential vaccine efficacy based on whether circulating viruses "matched" the vaccine product. Generally, I very much appreciated the authors' attention to detail and in-depth presentation of exactly what amino acids differ at what sites between vaccine strains and circulating viruses along with detailed presentation of genotype-by-VE results. However, I felt the authors did not go far enough in their analyses to properly support the primary conclusion of the paper that:

"The finding that CYD-TDV shows relatively consistent efficacy profiles against all genotypes of DENV-1 and -3 suggests that if any antigenic differences exist in these virus populations they are not measurably important to vaccine performance."

The authors’ logic is roughly as follows:

1) Measure overall AA difference between vaccine strains and circulating viruses in the ENV protein.

2) Also measure AA differences at relevant epitope sites between vaccine strains and circulating viruses (where "relevant epitope sites" are defined by monoclonals).

3) Find that overall AA similarity and AA similarity at epitope sites is high (>97%).

4) Separately find that there are no statistically significant differences in examining genotype-by-VE.

5) Conclude that within-serotype antigenic differences are not "measurably important to vaccine performance.

I agree with the authors that points 1-4 are useful avenues for investigating the hypothesis at hand (and I fully agree with the methods used to support points 1-4). However, I do not agree that points 1-4 are sufficient to reach the conclusion in point 5.

Regarding point 3, there are examples from influenza in which a single amino acid change reduces VE by ~50% (the 159Y mutation in H3N2 viruses in the 2014-2015 season). It is conceivable that VE would vary across dengue genotypes even if the dynamic range of AA similarity is between 97% and 98.5%.

Additionally, regarding point 4, estimates of genotype-specific VE often have a CI of >40% (Table 2). I realize this is dictated by the quantity of data available, but it is still an open question of whether there is statistical power to declare that there is no across-genotype differences in VE. To be convinced, I need a power calculation. What magnitude of across-genotype effect can the authors statistically reject?”

Required revisions for point A:

1) Authors should explore quantitatively the relationship between VE and AA difference, as for example here:http://www.cehd.umn.edu/ci/rationalnumberproject/images/90_3/figure_1.jpg

2) A strong (adequately powered) test of the null can provide confidence bounds on the degree of heterogeneity that include 0 and are narrow, which is not to say that the null is true but that a strong departure from the null is inconsistent with the data. A weak test of the null will provide wide bounds that may include 0 and also very large degrees of heterogeneity. The similarity of point estimates for different genotypes, alone, provides almost no information about the existence of heterogeneity or lack thereof. Therefore the authors need to provide some evidence about how strong their evidence is for near-heterogeneity of the VE – and based on the wide confidence intervals it seems not very strong for some genotypes. I can think of two ways to do this; the authors could choose one or perhaps take a different approach to provide equivalent information:a) A post-hoc power calculation that would show how large a difference in genotype-specific efficacy (or per-AA dropoff in efficacy) the sample is powered to detect.b) Alternatively, a statement of the confidence intervals for the interaction between vaccination and genotype (or vaccination and AA distance from the vaccine strain) that would measure the plausible amounts of heterogeneity observed.

Together with one of these or something similar, the phrasing in the Abstract and the Discussion about the likely homogeneity of effect should be changed to reflect the findings – again, based on the CI it seems the findings will likely be that the data are consistent with some or moderate heterogeneity for ST1, with no to very much heterogeneity for ST2 (underpowered), with considerable heterogeneity for ST4, and it's hard to say for ST3 as the 2 studies are well powered for different genotypes.

B) The manuscript should address in more detail the level of uncertainty in other findings, including the imputation.

C) The manuscript should more fully address the details of the reduced efficacy of the vaccine among seronegatives and younger children.

D) Little support is given for the particular epitopes considered in the analysis of divergence at mAb targets, except with references made in the legend of supplemental figures. Specifically, can the authors provide a rationale and/or reference data or literature to support the choice of the twelve epitope sequences that they state are "well characterised and potent virus neutralising human mAbs" (subsection “Human mAb epitope sequences in vaccine and wild-type viruses”).

*Reviewer #1:*

In this manuscript authors Rabaa and colleagues investigate the important issue of whether the CYD14 and CYD15 phase III dengue vaccine trials showed differential vaccine efficacy based on whether circulating viruses "matched" the vaccine product. Generally, I very much appreciated the authors' attention to detail and in-depth presentation of exactly what amino acids differ at what sites between vaccine strains and circulating viruses along with detailed presentation of genotype-by-VE results. However, I felt the authors did not go far enough in their analyses to properly support the primary conclusion of the paper that:

"The finding that CYD-TDV shows relatively consistent efficacy profiles against all genotypes of DENV-1 and -3 suggests that if any antigenic differences exist in these virus populations they are not measurably important to vaccine performance."

The authors’ logic is roughly as follows:

1) Measure overall AA difference between vaccine strains and circulating viruses in the ENV protein.

2) Also measure AA differences at relevant epitope sites between vaccine strains and circulating viruses (where "relevant epitope sites" are defined by monoclonals).

3) Find that overall AA similarity and AA similarity at epitope sites is high (>97%).

4) Separately find that there are no statistically significant differences in examining genotype-by-VE.

5) Conclude that within-serotype antigenic differences are not "measurably important to vaccine performance".

I agree with the authors that points 1-4 are useful avenues for investigating the hypothesis at hand (and I fully agree with the methods used to support points 1-4). However, I do not agree that points 1-4 are sufficient to reach the conclusion in point 5.

Regarding point 3, there are examples from influenza in which a single amino acid change reduces VE by ~50% (the 159Y mutation in H3N2 viruses in the 2014-2015 season). It is conceivable that VE would vary across dengue genotypes even if the dynamic range of AA similarity is between 97% and 98.5%.

Additionally, regarding point 4, estimates of genotype-specific VE often have a CI of >40% (Table 2). I realize this is dictated by the quantity of data available, but it is still an open question of whether there is statistical power to declare that there is no across-genotype differences in VE. To be convinced, I need a power calculation. What magnitude of across-genotype effect can the authors statistically reject?

Major suggestions for improvement to bridge to the conclusion at point 5:

1) It is fully possible that the authors' results can be explained with a consistent relationship between AA distance between vaccine strain and genotype and genotype-specific vaccine efficacy while not showing an obvious difference of VE across individual genotypes. I'm imagining a situation like shown here:

http://www.cehd.umn.edu/ci/rationalnumberproject/images/90_3/figure_1.jpg

One sees a consistent relationship between the x variable (distance) and the y variable (VE) even if group-specific y values are hard to pin down.

I would strongly encourage the authors to do a similar analysis. Is there an overall correlation between amino acid distance (overall and at epitope sites) and mean estimates of genotype-specific VE?

2) The authors' primary conclusion needs to be justified with a power calculation. With the quantity of data that the authors have (6 cases and 6683 person-years-at-risk in the vaccine group of serotype 1 / genotype 1 in CYD14, etc.) what level of relationship between AA distance and VE do the authors have statistical power to perceive? This could be approached by generating simulated datasets with a specified effect size and then testing whether the statistical approach has power to detect this effect size.

This would allow the authors to reach the more justified and defined conclusion that within-serotype antigenic differences must be less than X in magnitude rather than antigenic differences are not "measurably important to vaccine performance".

Without these additional analyses, I do believe the paper's major conclusions can be supported.

*Reviewer #2:*

In this manuscript, Rabaa and coathors demonstrate pan-genotype and pan-serotype efficacy for the CYD DENV vaccine by generating 454 sequenced data for 314 and 333 viral E genes from patient samples deriving from the CYD14 and CYD15 phase 3 clinical trials, respectively. The samples derive from children with a range of ages, hailing from eleven countries, and the authors examine genotype-specific efficacy for 10 clades of the E gene.

This is a well-written and largely clearly presented manuscript. The authors acknowledge the two most significant weaknesses of the analysis: the strong and well-documented bias with regard to generation for samples with high viral titer, and the power limitations of sample size for the calculation of genotype-specific efficacy given the large number of serotypes and genotypic clades examined (not to mention the significant genetic diversity within each genotypic clade).

The authors could address the former issue more directly by looking for association between successfully amplified viral genotypes and viral titer; if there are no trends or genotype associations within the genotype-able range of titer, that would be reassuring.

The latter issue is more difficult to address with the available data. Larger sample size with less stratification would be desirable. The authors describe part of the motivation for this study as a desire to determine whether a vaccine 'escape response' could result from genotype-specific efficacy. Are there models or data to suggest what magnitude of genotype-specific efficacy could induce an escape response in a DENV population, and if so, does the present study have power to detect selective pressures of that relevant magnitude?

Specific points:

Subsection “Sequence differences between CYD-TDV vaccine strains and circulating wild-type viruses” and elsewhere: What does X% sequence divergence mean? How large is the E gene construct in the CYD vaccine, and how many amino acid positions were different in the observed samples?

Subsection “Vaccine efficacy by DENV serotype and genotype”, second paragraph: It would be useful in this discussion of genotype-specific efficacy to mention which genotype clades are represented in the CYD vaccine. Otherwise, it is necessary to manually annotate Table 1 and Table 2 after referencing and interpreting the phylogenetic tree figures.

Subsection “Data Analysis”: Was there any validation of genotype/sequencing accuracy performed? Even if phred scores were high, were there issues with homopolymeric sequences, a notorious weak point of 454 sequencing? Were real or spurious insertions or deletions observed? Were any technical replicates performed on samples to evaluate data via consistency?

*Reviewer #3:*

The manuscript "Genetic epidemiology of dengue virus populations in phase III trials of the CYD tetravalent dengue vaccine and implications for efficacy" presents an analysis of dengue viruses that were detected during a vaccine trial. The study characterizes dengue viruses that were isolated from vaccinated and individuals who experienced infection during the trial. The study compares parental viruses of the vaccine to circulating viruses and estimates genotype specific efficacy rates. The study provides important information about the performance of the vaccine as well as interesting virus sequences (for interests even independent of the vaccine trial) in providing a snapshot of dengue viruses circulating in multiple locations at the same time.

The manuscript makes an important contribution to the field. My main concerns are that 1) little support is given for the particular epitopes considered, except with references made in the legend of supplemental Figure 2) consideration of uncertainty in each of the analyses is insufficient including the imputation and the details of the reduced efficacy of the vaccine among seronegatives and younger children should be an additional focus of the manuscript. Overall, the strengths of the manuscript outweigh the weaknesses.

[Editors' note: further revisions were requested prior to acceptance, as described below.]

Thank you for resubmitting your work entitled "Genetic epidemiology of dengue virus in phase III trials of the CYD tetravalent vaccine and implications for efficacy" for further consideration at *eLife*. Your revised article has been evaluated by Wendy Garrett (Senior Editor) and a Reviewing Editor.

The manuscript has been improved but there are some remaining issues that need to be addressed before acceptance, as outlined below:

From the Reviewing Editor:

I remain concerned that underpowered and thus uninformative analyses are presented as results. The qualifications stated in the revised version help this somewhat, but the paper continues to give the impression of finding that efficacy was equivalent between genotypes except for DENV4.

The comment in the decision letter was that the authors need to perform either post-hoc power calculations or report confidence intervals for the evidence for interaction, consistent with the very strong recent statement from the American Statistical Association that p values alone are not a grounds for scientific conclusion, and recommendation: "In view of the prevalent misuses of and misconceptions concerning p-values, some statisticians prefer to supplement or even replace p-values with other approaches. These include methods that emphasize estimation over testing, such as confidence, credibility, or prediction intervals; Bayesian methods; alternative measures of evidence, such as likelihood ratios or Bayes Factors; and other approaches such as decision-theoretic modeling and false discovery rates. All these measures and approaches rely on further assumptions, but they may more directly address the size of an effect (and its associated uncertainty) or whether the hypothesis is correct." (http://amstat.tandfonline.com/doi/abs/10.1080/00031305.2016.1154108)

The authors declined a post-hoc power calculation on the advice of a statistician, but did not address the other suggestion to calculate confidence intervals for the interactions, which would be trivial to do. In the case of DENV2 in particular, it seems that the calculation of these intervals would lay bare that the study simply offers little evidence on the question, rather than as the authors say a difference that was not statistically significant.

I repeat this request for CI for the interactions and believe it needs to be addressed before acceptance. As an example of what I believe this will show: The statement in the Abstract currently is:

"In post-hoc analysis of all CYD14/15 trial participants, the only statistically significant genotype-level VE association was within DENV-4, where efficacy against genotype I was lower than to genotype II. In trial participants age 9-16 years, no genotype-level associations with VE were observed, although this subgroup analysis had less precision due to smaller sample sizes."

I suspect that the confidence bounds for interaction will be tight for DENV-4 and exclude 0: Thus for DENV-4, the authors can conclude that efficacy against genotype I was lower than for Genotype II.

I expect the confidence bounds for interaction will be extremely wide for DENV-2, including both zero and very large differences: if true this means the study results gave little information about efficacy differences between genotypes.

I expect the confidence bounds for interaction will be relatively tight around DENV 1, indicating that the study provides evidence that if there is heterogeneity, it is relatively small.

For DENV3 I expect the results to be intermediate between the cases for DENV1 and DENV2, meaning the study results are consistent with heterogeneity from zero to moderate.

These are the legitimate conclusions. Stating a trend, but noting lack of significance and small sample size clouds the issue. The goal of a study like this is to narrow the possible states of the world consistent with the data. For DENV4 that has been done to strongly suggest heterogeneity exists and may be substantial. For DENV1 that has been done to strongly suggest that heterogeneity is absent or at most small. For DENV2 that has not been done to an appreciable extent, and DENV3 is in between.

---

## [Author Response]

*[…] Required revisions for point A:*

*1) Authors should explore quantitatively the relationship between VE and AA difference, as for example here:http://www.cehd.umn.edu/ci/rationalnumberproject/images/90_3/figure_1.jpg*

As suggested, we have generated a supplementary figure (Figure 3—figure supplement 1) to represent the relationship between VE and AA identity (between the individual vaccine strains and the wild-type viruses sampled in the trial). This figure is illustrative only and the confidence intervals around the VE estimates preclude any attempt at correlation analyses. This approach nonetheless provides a useful precedent to the field in how to present such data in future dengue vaccine trials.

*2) A strong (adequately powered) test of the null can provide confidence bounds on the degree of heterogeneity that include 0 and are narrow, which is not to say that the null is true but that a strong departure from the null is inconsistent with the data. A weak test of the null will provide wide bounds that may include 0 and also very large degrees of heterogeneity. The similarity of point estimates for different genotypes, alone, provides almost no information about the existence of heterogeneity or lack thereof. Therefore the authors need to provide some evidence about how strong their evidence is for near-heterogeneity of the VE – and based on the wide confidence intervals it seems not very strong for some genotypes. I can think of two ways to do this; the authors could choose one or perhaps take a different approach to provide equivalent information:a) A post-hoc power calculation that would show how large a difference in genotype-specific efficacy (or per-AA dropoff in efficacy) the sample is powered to detect.b) Alternatively, a statement of the confidence intervals for the interaction between vaccination and genotype (or vaccination and AA distance from the vaccine strain) that would measure the plausible amounts of heterogeneity observed.*

*Together with one of these or something similar, the phrasing in the Abstract and the Discussion about the likely homogeneity of effect should be changed to reflect the findings – again, based on the CI it seems the findings will likely be that the data are consistent with some or moderate heterogeneity for ST1, with no to very much heterogeneity for ST2 (underpowered), with considerable heterogeneity for ST4, and it's hard to say for ST3 as the 2 studies are well powered for different genotypes.*

The main critique is that we have not adequately qualified our interpretation of our finding that vaccine efficacy did not differ between genotypes of the same serotype in the trial data. We agree that the absence of a measurable association does not prove the null, i.e. that there is no true difference in VE between genotypes. With respect to the suggestion that we perform post hoc power calculations – we consulted with independent statisticians and there is clearly not a consensus opinion; indeed the literature suggests post hoc power calculations can have significant drawbacks and may imply greater certainty around these estimates (1, 2). Our response therefore is to modify the manuscript text to be appropriately more conservative in our interpretation. In the revised manuscript, we have placed greater focus on the uncertainty in the point estimates of VE, revealed by the size of the 95% CIs, and repeatedly described the uncertainty inherent to VE estimates from sometimes small sample sizes.

Line by line edits to the manuscript addressing this critique and resulting in a more balanced perspective of the results:

1) Abstract: We removed text suggesting that our findings were broadly reassuring that efficacy would not be influenced by within-serotype genetic diversity, instead we simply state our observations relating to DENV-4 and qualify the subgroup analysis herewith “In post hoc analysis of all CYD14/15 trial participants, the only statistically significant genotype-level VE association was within DENV-4, where efficacy against genotype I was lower than to genotype II. In trial participants aged 9-16 years, no genotype-level associations with VE were observed, although this subgroup analysis had less precision due to smaller sample sizes. Post-licensure surveillance is needed to monitor vaccine performance against the backdrop of DENV sequence diversity and evolution.”

2) Introduction, “Lastly, we aimed to explore if a more complex genotype-specific efficacy pattern existed in the CYD14 and CYD15 trials, notwithstanding the limitations inherent in post hoc analysis.”

3) Results, “However, when the genotype-specific efficacy was considered only in participants ≥ 9 years, DENV-4 vaccine efficacy was comparable across all groups (80.8-89.6%) and no significant heterogeneity was detected between genotypes (p=0.67), although the sample sizes were small and the confidence intervals around the point estimates wide (Table 2).”

4) Discussion, “That CYD-TDV immunisation induced similar measured levels of efficacy against all genotypes of DENV-1 and -3 is so far supportive of the concept that clinical immunity elicited by CYD-TDV to these serotypes was pan-genotype in nature (Table 1 and Table 2). Nonetheless further research is warranted to understand if smaller differences exist in genotype-level vaccine efficacy than could be measured in these trials and that might be relevant to programmatic use of vaccine.”

5) Discussion, “Subgroup analyses of genotype and age-stratified vaccine efficacy are inevitably speculative because of diminishing sample sizes and wide confidence intervals around the point estimates.”

6) Discussion, “As above for DENV-4, it is speculative to examine subgroups, but for the age group 9-16 years of age, the age range eligible for the now licensed vaccine, the efficacy against DENV-2 Asian I genotype (34.6% (-27.4; 65.7) was comparable to that seen against the other DENV-2 genotypes (Asian/American and Cosmopolitan), albeit with inevitably wide confidence intervals around the point estimates.”

7) Discussion, “These were post hoc analyses and inevitably sample sizes became small for genotype level vaccine efficacy estimates and in particular the age-class (9-16 year olds) subgroup analysis. This manifests as wide 95% confidence intervals around the genotype-level point estimates of vaccine efficacy.”

8) Discussion, “Despite these limitations, with the exception of DENV-4, the results described here do not reveal large and significant differences in vaccine efficacy between genotypes of the same serotype in the CYD14/15 trials. However, these data do not preclude the possibility that a true difference exists, just that a larger sample size than that available in this study would be required to detect this.”

9) Discussion: We removed text suggesting that the lack of heterogeneity in efficacy estimates is an indication that efficacy is homogenous across diverse genotypes, replacing it with a more clear statement of our findings only, that “the results described here do not reveal a significant difference in VE between genotypes of the same serotype in the CYD14/15 trials. However these data do not preclude the possibility that a true difference exists, just that a larger sample size than that available in this study would be required.”

*B) The manuscript should address in more detail the level of uncertainty in other findings, including the imputation.*

The critique suggests there is uncertainty in the imputation of genotype identity. We recognize some uncertainty and have acknowledged this in the manuscript. However, as described in the manuscript, we believe that uncertainty in the imputation is low overall and particularly so for Latin America. In all of the CYD15 Latin American countries, not only did we identify just one genotype per serotype in each country (with the exception of a single isolate of genotype I DENV-4, a likely importation from Southeast Asia), but the virus genotype imputation was entirely supported by contemporary independent descriptions of the local DENV population genetics, i.e. the genotype identity of ~200 DENV sequences submitted independently to Genbank between 2010-2014 by other investigators were identical to the country-level genotype assignments reported for the CYD15 patients in this study, and thus supported our imputation approach (Supplementary file 3).

For CYD14, Supplementary file 3 provides genotype identity to viruses submitted to Genbank between 2010 and 2014. In all instances the genotype we imputed for the country level data CYD14 results was the majority sequence genotype in that country, thus generally supporting the validity of the imputation methods we used for missing data CYD14 genotype data.

*C) The manuscript should more fully address the details of the reduced efficacy of the vaccine among seronegatives and younger children.*

It is not possible to do any meaningful virus genotype analyses in the “sero-cohort” because these presented only 10% of the total number of participants (see the respective CYD trial manuscripts). We note our inability to assess this due to small sample sizes in the last paragraph of the Discussion. For the benefit of readers, we show the genotype level vaccine efficacy results for children under 9yrs of age in Supplementary file 5 and make mention of this in the Results text. We do not elaborate greatly on these results because only the CYD14 trial enrolled participants in this age range and thus the numbers of cases per genotype is small and inevitably this yields very large 95% CI around the efficacy point estimates.

*D) Little support is given for the particular epitopes considered in the analysis of divergence at mAb targets, except with references made in the legend of supplemental figures. Specifically, can the authors provide a rationale and/or reference data or literature to support the choice of the twelve epitope sequences that they state are "well characterised and potent virus neutralising human mAbs" (subsection “Human mAb epitope sequences in vaccine and wild-type viruses”).*

We have now added the appropriate references for the mAbs in the Results section in addition to their original referencing in the Materials and methods section. We have also added text to explain why these represent prominent neutralizing mAb epitopes under investigation by the dengue vaccine community.

References:

1) J. M. Hoenig, D. M. Heisey, The abuse of power: the pervasive fallacy of power calculations for data analysis, Am. Stat. 55, 19–24 (2001).

2) R. V Lenth, Post hoc power: tables and commentary, Iowa City Dep. Stat. Actuar. Sci. Univ. Iowa (2007).

[Editors' note: further revisions were requested prior to acceptance, as described below.]

*[…] These are the legitimate conclusions. Stating a trend, but noting lack of significance and small sample size clouds the issue. The goal of a study like this is to narrow the possible states of the world consistent with the data. For DENV4 that has been done to strongly suggest heterogeneity exists and may be substantial. For DENV1 that has been done to strongly suggest that heterogeneity is absent or at most small. For DENV2 that has not been done to an appreciable extent, and DENV3 is in between.*

We appreciate the reviewers’ concern for our assessment of uncertainty and the language that they used to push us in the right direction. As suggested, we have now adjusted our analyses to include estimates of the interaction between vaccine efficacy and genotype, along with 95% confidence intervals defining the uncertainty around these estimates. We have added this to the Materials and methods, which now indicate “A Cox regression model was used to estimate vaccine efficacy (derived as 100* [1- Hazard Ratio]) with vaccine group included as a covariate and 95% CI. To further investigate the interaction between vaccine efficacy and genotype, an additional Cox regression model was used to estimate vaccine efficacy with vaccine group, genotype and the interaction between vaccine group and genotype included as covariates.”

We have added two tables with these results, Table 3 (all age groups) and Table 4 years of age). The results included in these tables are generally as expected by the reviewer. For DENV-4, they confirm the decreased efficacy of genotype I in the all age group, and confirm a lack of heterogeneity in the ≥ 9 year olds. For DENV-1, tight confidence intervals suggest limited uncertainty around our conclusion of little to no heterogeneity. For DENV-3, wide confidence intervals suggest that we have limited power to assess heterogeneity in vaccine efficacy in these groups. In the case, of DENV-2, we interestingly find confidence bounds including, but very close to zero, in comparison of the Asian I genotype to the American/Asian genotype in the all ages group, which suggests that although our power to detect a difference is limited, there may be an interaction that we are just beyond the limit of detecting. We have added these results to the manuscript in the Results section (subsection “Vaccine efficacy by DENV serotype and genotype”, last four paragraphs) and in the Discussion (fourth, fifth and sixth paragraphs).

To clarify this uncertainty further, we have also altered the Abstract to read: “Post-hoc analysis of all CYD14/15 trial participants revealed a statistically significant genotype-level VE association within DENV-4, where efficacy was lowest against genotype I. In subgroup analysis of trial participants age 9-16 years, VE estimates appeared more balanced within each serotype, suggesting that genotype-level heterogeneity may be limited in older children.”